# FINITE-TIME CONVERGENCE ANALYSIS OF ODE-BASED GENERATIVE MODELS FOR STOCHASTIC INTERPOLANTS

**Yuhao Liu   Yu Chen   Rui Hu   Longbo Huang**[*]
IIIS, Tsinghua University
`{liuyuhao21,chenyu23,hu-r24}@mails.tsinghua.edu.cn`
`longbohuang@tsinghua.edu.cn`

## ABSTRACT

Stochastic interpolants offer a robust framework for continuously transforming samples between arbitrary data distributions via ordinary or stochastic differential equations (ODEs/SDEs), holding significant promise for generative modeling. While previous studies have analyzed the finite-time convergence rate of discrete-time implementations for SDEs, the ODE counterpart remains largely unexplored. In this work, we bridge this gap by presenting a rigorous finite-time convergence analysis of numerical implementations for ODEs in the framework of stochastic interpolants. We establish novel discrete-time total variation error bounds for two widely used numerical solvers: the first-order forward Euler method and the second-order Heun's method. Our analysis also yields optimized iteration complexity results and step size schedules that enhance computational efficiency. Notably, when specialized to the diffusion model setting, our theoretical guarantees for the second-order method improve upon prior results in terms of both smoothness requirements and dimensional dependence. Our theoretical findings are corroborated by numerical and image generation experiments, which validate the derived error bounds and complexity analyses.

## 1 INTRODUCTION

Stochastic interpolants (Albergo & Vanden-Eijnden, 2023; Albergo et al., 2023) provide a powerful framework for constructing generative models by learning deterministic or stochastic transformations that continuously map samples from an initial distribution $\rho_0$ to a target distribution $\rho_1$, governed by ordinary or stochastic differential equations (ODEs/SDEs). To determine the differential equation, the approach constructs stochastic interpolations between $\rho_0$ and $\rho_1$ samples, then estimates a mean velocity field from these paths. With a learned approximation of the velocity field, one can build a generative model by solving the approximated ODE or SDE. This novel framework unifies flow matching (Lipman et al., 2023) and score-based diffusion (Song & Ermon, 2020; Ho et al., 2020; Song et al., 2021), offering significant design flexibility through its various choices of the initial distribution and generalized interpolation formulation, making it an important subject for generative modeling (Chen et al., 2024b; Albergo et al., 2024a;b; Chen et al., 2024a).

Recent theoretical work has focused on understanding the convergence properties of stochastic interpolants. For transformations based on stochastic differential equations (SDEs), theoretical guarantees on generation error bounds have been established for both continuous-time (Albergo et al., 2023) and discrete-time (Liu et al., 2025) scenarios. Continuous-time analysis focuses on the exact solutions of the underlying equations, while discrete-time analysis further considers the numerical approximation used in practice since the equations usually cannot be analytically solved. However, for their deterministic counterparts, ODE-based transformations, the analysis has been limited to the continuous-time setting (Albergo & Vanden-Eijnden, 2023; Benton et al., 2024b), despite their widespread use in practical applications. Therefore, we address this critical gap by investigating

---

[*]Corresponding author.

the following question: **What is the iteration complexity for numerical ODE solvers to ensure accurate generation with stochastic interpolants?**

This problem is important yet technically challenging. Although there have been insightful analyses for diffusion models and related approaches that can transform Gaussian to target distributions (e.g., Li et al. 2025b; 2024b; Huang et al. 2025), they cannot be directly extended to stochastic interpolants due to the general data-to-data transformation structures which make the interpolant non-Markovian. While Liu et al. (2025) recently established finite-time error bounds for the SDE case, their methods face inherent challenges in ODE cases due to the singular behavior in the degenerated process.

This work bridges the gaps by presenting the first systematic analysis of discretization schemes for stochastic interpolant ODEs. Specifically, we provide a complete theoretical characterization of both first-order (forward Euler) and second-order (Heun's) methods applied to stochastic interpolants (Theorem 4.5 and 5.4). We explicitly quantified the discretization error introduced by numerical solvers under mild assumptions, and then derive $O(\varepsilon^{-1}d^2)$ and $O(\varepsilon^{-1/2}d^{3/2})$ iteration complexity bounds, respectively, for achieving $\varepsilon$ distribution approximation error on $d$-dimensional data.

A key technical innovation is the introduction of carefully designed continuous-time interpolations. These interpolations reformulate discrete-time iterations into ODE forms comparable to the true process, which enables the use of continuous-time analytical tools to quantify the approximation error by comparing the induced drifts and divergences with those of the true process.

Notably, our results establish a general analysis framework valid for every stochastic interpolant process satisfying mild regularity assumptions, including the famous variance-preserving diffusion (Song et al., 2021). Furthermore, our dedicated analysis on the Heun's method achieves state-of-the-art iteration complexity results on diffusion models (see Section 5.2 and Appendix B). This advancement arises from our detailed error analysis, where we carefully partition the drift and divergence errors into tractable expectation terms, yielding an improved error bound (check Appendix B.1 for further comparison).

The contributions of this paper are summarized below:

- We provide the first finite-time analysis for stochastic interpolant ODEs. We derive general TV error bounds for both first-order (Euler's) and second-order (Heun's) methods, which are applicable to all stochastic interpolant process satisfying mild regularity assumptions.

- We explicitly quantify the discretization error introduced by numerical ODE solvers, establishing $O(\varepsilon^{-1}d^2)$ and $O(\varepsilon^{-1/2}d^{3/2})$ iteration complexity bounds for both methods, respectively. Especially, when degenerating to the variance-preserving diffusion model, our result for Heun's method improves the existing analyses on diffusion models.

- We employ carefully designed continuous-time interpolation to unify the comparison between the discrete-time approximation and continuous-time target processes, allowing us to quantify the approximation error by comparing the induced drifts and divergences. We also corroborate our findings with numerical experiments and validations on image generation.

## 2 RELATED WORKS

### 2.1 STOCHASTIC INTERPOLANTS ANALYSIS

The stochastic interpolant framework originates from continuous-time normalizing flows, providing a principled approach for constructing data-to-data generative models (Albergo & Vanden-Eijnden, 2023). Subsequent work (Albergo et al., 2023) extended this framework through the incorporation of Gaussian perturbations, enabling learnable score functions and facilitating the application of stochastic differential equations for data transformation tasks.

Regarding ODE-based formulations, Albergo & Vanden-Eijnden (2023) established Wasserstein error bounds for velocity field estimation under Lipschitz continuity assumptions. Benton et al. (2024b) advanced these results by considering time-dependent Lipschitz constants, thereby obtaining tighter error bounds. Their analysis further examined the control of Lipschitz constants specifically for linear interpolants. Both studies focused exclusively on continuous-time settings.

In the SDE context, Albergo et al. (2023) derived continuous-time Kullback-Leibler (KL) divergence bounds between target and estimated distributions, expressed in terms of the mean squared error of drift estimation. Liu et al. (2025) made progress by establishing finite-time error bounds for the Euler-Maruyama discretization scheme, representing the first discrete-time analysis within the stochastic interpolant framework. They also investigated the impact of schedule selection.

## 2.2 ODE-BASED DIFFUSION MODELS ANALYSIS

Recent theoretical work has made significant progress in analyzing the convergence properties of probability-flow ODE diffusion models. Chen et al. (2023b) provided the first polynomial convergence bounds for DDIM-type samplers. Chen et al. (2023a) developed a provably efficient sampling algorithm incorporating corrector steps within the ODE solver framework, though this approach introduces additional stochasticity to the process.

Recent theoretical analyses of purely deterministic generation typically adopt two principal approaches. The first directly examines discrete-time density evolution, where Li et al. (2024b) established a foundational framework for controlling the total variation (TV) distance between target and estimated distributions. Subsequent works (Li et al., 2024a;c; 2025a; Liang et al., 2025) extended this framework to derive tighter bounds and accelerated convergence guarantees. The second approach considers equivalent continuous-time processes through partial differential equation (PDE) analysis. While both Huang et al. (2025) and (Li et al., 2025b) employed PDE techniques to bound TV error, the latter achieved superior results through more refined error decomposition methods.

Current theoretical understanding indicates that first-order methods only require $\tilde{O}(d/\varepsilon)$ iterations to achieve $\varepsilon$-precision in TV distance (Li et al., 2024c). However, existing analyses of higher-order methods remain either non-tight or reliant on overly restrictive assumptions, highlighting the need for more comprehensive theoretical frameworks for ODE-based approaches. See Appendix B and Table 1 for a comparison between our results and existing works.

## 3 BACKGROUND ON STOCHASTIC INTERPOLANTS

Consider two probability distributions $\rho_0, \rho_1 \in \Delta(\mathbb{R}^d)$. The stochastic interpolant framework (Albergo & Vanden-Eijnden, 2023; Albergo et al., 2023) provides a general approach for constructing a learnable mapping between $\rho_0$ and $\rho_1$: by identifying a learnable vector field $b(t, x)$ such that the solution of ODE (see Appendix A for SDE cases)

$$\mathrm{d}X_t = b(t, X_t)\mathrm{d}t \qquad (*)$$

satisfies $X_1 \sim \rho_1$ under initial condition $X_0 \sim \rho_0$.

Specifically, $b(t, x)$ is defined as the average velocity $b(t, x) = \mathbb{E}[\dot{x}_t | x_t = x]$ of a *stochastic interpolant* $(x_t)_{t \in [0,1]}$ between $\rho_0$ and $\rho_1$, given by:

$$x_t := I(t, x_0, x_1) + \gamma(t)z,$$

where

- $(x_0, x_1) \sim \nu$ with marginals $x_0 \sim \rho_0$, $x_1 \sim \rho_1$, and $z \sim \mathcal{N}(0, I_d)$ is an independent noise.
- $I$ is a $C^2$-smooth interpolation satisfying $I(0, x_0, x_1) = x_0$ and $I(1, x_0, x_1) = x_1$.
- $\gamma(t) > 0$ for $t \in (0, 1)$, so $\gamma(t)z$ serves to smooth the marginal distributions.

Notably, the coupling $\nu$ can handle both paired and unpaired data, given $(x_0, x_1)$ are dependent or independent, respectively. Let $\rho(t)$ denote the law of $x_t$ and $\rho(t, \cdot)$ its density. Typically, we choose $\gamma(0) = \gamma(1) = 0$ so that $\rho(0) = \rho_0$ and $\rho(1) = \rho_1$.

Under this definition of $b(t, x)$, the solution $X_t$ of ODE $(*)$ satisfies $X_t \sim \rho(t)$ for $t \in [0, 1]$. In practice, since $b(t, x)$ is generally not analytically computable, we usually use neural networks to obtain an estimate $\hat{b}(t, x)$ (see Appendix A for more details). By applying numerical methods to solve the ODE $(*)$ with $\hat{b}(t, x)$, we can then transform samples from $\rho_0$ to $\rho_1$.

**Remark:** The widely used diffusion model can be viewed as a special case of stochastic interpolants. For example, consider the linear interpolant $x_t = (1-t)x_0 + tx_1 + \sqrt{2t(1-t)}z$ with $x_0 \sim \rho_0 =$

$\mathcal{N}(0, I_d)$ and $x_1 \sim p_{\text{data}}$. In this case, the marginal distributions of $x_t$ coincide with those of the variance-preserving (VP) diffusion model (see, e.g., Song et al. 2021).

Below, we conduct a novel analysis for stochastic interpolants based on two foundational methods: the Euler's method (Section 4) and the Heun's method (Section 5). We derive finite-time TV error bounds for both methods (Theorem 4.5 and 5.4) respectively, and compare their convergence rates.

## 4 THE FIRST-ORDER RESULTS FROM THE FORWARD EULER METHOD

We start by analyzing the ODE-based stochastic interpolants with the usage of the forward Euler method, an important first-order scheme for solving ordinary differential equations. We derive a finite-time error bound for its numerical solution, which further allows us to analyze the iteration complexity.

### 4.1 ERROR BOUND FOR THE FORWARD EULER METHOD

To begin with, we formally define the iterations of the forward Euler method as follows. Consider a time discretization given by the schedule $\{t_k\}_{k=0}^{N}$, where $t_0 < t_1 < t_2 < \cdots < t_N$. Given an initial condition $\hat{X}_{t_0}$, the forward Euler method approximates the solution of ODE $(*)$ iteratively by $\{\hat{X}_{t_i}\}_{i=0}^{N}$. Specifically, at iteration $(k+1)$, we set

$$\hat{X}_{t_{k+1}} := \hat{X}_{t_k} + h_k \cdot \hat{b}(t_k, \hat{X}_{t_k}), \tag{Euler}$$

where $\hat{b}(t, x)$ is an estimator of the drift function $b(t, x)$, and we use $h_k = t_{k+1} - t_k$ to denote the step size. Our objective is to quantify and control the difference between the approximated terminal distribution $\hat{\rho}(t_N)$ given by $\hat{X}_{t_N}$ and the true terminal distribution $\rho(t_N)$.

We make the following assumptions to rigorously analyze the forward Euler method.

**Assumption 4.1** (Regularity). *Let $\nu$ be the joint distribution of $\rho_0$ and $\rho_1$, we assume that:*

- *The fourth moment of the difference between the initial and goal distribution is bounded:* $\mathbb{E}_{(x_0, x_1) \sim \nu} \left[ \|x_0 - x_1\|^4 \right] < \infty.$

- *The $p$-th order time derivatives (for $p = 1, 2$) of $I(t, x_0, x_1)$ and $\gamma^2(t)$ are bounded w.r.t. $t$: There exist positive constants $C_I, C_\gamma > 0$ such that for all $x_0, x_1 \in \mathbb{R}^d$ and $p \in \{1, 2\}$,*

$$\|\partial_t^p I(t, x_0, x_1)\| \leq C_I \|x_0 - x_1\|, \quad \left| \frac{\mathrm{d}^p}{\mathrm{d}t^p} \left[ \gamma^2(t) \right] \right| \leq C_\gamma.$$

Assumption 4.1 ensures the regularity of both the stochastic interpolant process $(x_t)_{t \in [0,1]}$ and the function $b(t, x)$. For example, with the linear interpolant $x_t = (1 - t)x_0 + tx_1 + \sqrt{2t(1 - t)}z$, this assumption is met if $\rho_0$ and $\rho_1$ both have finite fourth moments.

**Assumption 4.2** (Drift Estimation Error). *Let $\varepsilon_{1,k}(x) = \|\hat{b}(t_k, x) - b(t_k, x)\|$ represent the error between the estimated drift coefficient $\hat{b}(t_k, x)$ and the true drift coefficient $b(t_k, x)$ at time $t_k$ and state $x$. We assume that*

$$\sum_{k=0}^{N-1} h_k \mathop{\mathbb{E}}_{x_{t_k} \sim \rho(t_k)} \left[ \varepsilon_{1,k}(x_{t_k})^2 \right] \leq \varepsilon_{drift}^2 < \infty.$$

Assumption 4.2 is a standard condition for estimator quality, similar to those in (Benton et al., 2024a; Liu et al., 2025). This assumption differs from continuous-time conditions by providing a discrete-time formulation that focuses on the error at the time steps used in the numerical method.

However, unlike SDE-based models, Assumption 4.2 alone is insufficient to fully control the distribution error (Li et al., 2024b;a; 2025b). Therefore, we introduce further assumptions on $\hat{b}(t, x)$.

**Assumption 4.3** (Jacobian Estimation Error). *Let $\varepsilon_{2,k}(x) = \left\| \nabla\hat{b}(t_k, x) - \nabla b(t_k, x) \right\|_F$, we assume*

$$\sum_{k=0}^{N-1} h_k \mathop{\mathbb{E}}_{x_{t_k} \sim \rho(t_k)} \left[ \varepsilon_{2,k}(x_{t_k}) \right] \leq \varepsilon_{div} < \infty.$$

**Assumption 4.4** (Smoothness). *$\hat{b}(t, x)$ is $C^2$ w.r.t. $x$. Furthermore, there is a constant $L > 0$ such that for all $k = 0, 1, \ldots, N$ and all $x, y \in \mathbb{R}^d$,*

$$\left\| \nabla\hat{b}(t_k, x) \right\|_F \leq L, \ \left\| \nabla^2\hat{b}(t_k, x) \right\|_F \leq L^{3/2}, \ \|\nabla \cdot \hat{b}(t, x) - \nabla \cdot \hat{b}(t, y)\| \leq L^{3/2}\|x - y\|.$$

Assumption 4.3 extends the estimation accuracy requirement of Assumption 4.2 to the Jacobian of $\hat{b}(t, x)$. Similar assumptions are used by Li et al. (2024a) and Li et al. (2025b). Assumption 4.4 imposes a uniform Lipschitz constant on both $\hat{b}(t, x)$ and its divergence. Its reasonableness is illustrated by cases where the data for $\rho_0$ and $\rho_1$ are bounded in each dimension; in such cases, Lemma F.7 in the Appendix shows that $L$ is of order $O(d)$.

With the preceding assumptions established, we are now ready to present the main theoretical result concerning the application of the forward Euler method.

**Theorem 4.5.** *Under Assumptions 4.1, 4.2, 4.3 and 4.4, suppose the forward Euler method is initialized with $\hat{X}_{t_0} \sim \hat{\rho}(t_0)$, and the step sizes satisfy $h_k \leq \frac{1}{2L}$. Then,*

$$\mathrm{TV}(\rho(t_N), \hat{\rho}(t_N)) \lesssim \mathrm{TV}(\rho(t_0), \hat{\rho}(t_0)) + \varepsilon_{div} + \varepsilon_{drift} \left( d^{1/2} S(\gamma, t_0, t_N)^{1/2} + L^{1/2} \right)$$

$$+ \underbrace{\sum_{k=0}^{N-1} h_k^2 \left[ \bar{\gamma}_k^{-4} d^2 + \bar{\gamma}_k^{-2} M^2 \right]}_{\text{Discretization Error}}.$$

*Here, we have defined: $\bar{\gamma}_k := \inf_{t \in [t_k, t_{k+1}]} \gamma(t)$, $S(\gamma, t_0, t_N) := \int_{t_0}^{t_N} \gamma^{-2}(t)\mathrm{d}t$ and $M := \max\left\{ d, L, \sqrt{\mathbb{E}_\nu \left[ \|x_0 - x_1\|^4 \right]} \right\}$.*

Theorem 4.5 provides a comprehensive upper bound on the total variation (TV) distance between the true target distribution $\rho(t_N)$ and its approximation $\hat{\rho}(t_N)$ obtained via the forward Euler method. This result quantifies the convergence behavior of the Euler scheme, and elucidates how several factors influence the approximation error: the initialization error $\mathrm{TV}(\rho(t_0), \hat{\rho}(t_N))$, dimension $d$, the distance between source and target distribution captured by $\mathbb{E}_\nu[\|x_0 - x_1\|^4]$, the Lipschitz constant $L$, the estimation errors $\varepsilon_{drift}$ and $\varepsilon_{div}$, and the step size schedule $\{h_k\}_{k=0}^{N-1}$.

The step sizes $\{h_k\}_{k=0}^{N-1}$ specifically influence the last term **discretization error** in Theorem 4.5, which quantifies the influence of using the discrete-time numerical solver. The other terms are independent with the step size schedule, reflecting the inherent quality achievable with an infinite number of steps. Since the discretization error term is of order $O(h_k^2)$ for each step, summing them up will yield an error bound proportional to the step size (see Section 4.2 for a detailed analysis when the schedule is specified), so it approaches zero as the step sizes tend to zero. In addition, the $O(h_k^2)$ discretization error per step coincides with prior works for diffusion models with first-order samplers (Li et al. 2025b; Huang et al. 2025).

To avoid an unbounded right-hand side in Theorem 4.5 when $\gamma(0) = 0$ or $\gamma(1) = 0$ (which ensures $\rho(0) = \rho_0$ and $\rho(1) = \rho_1$ in the stochastic interpolant definition), we follow Liu et al. (2025) and simulate the process within a sub-interval $[t_0, t_N] \subset (0, 1)$. This means our sampling starts from an estimation of $\rho(t_0)$ rather than $\rho_0$, and similarly aims for an estimation of $\rho(t_N)$ instead of $\rho_1$. When $t$ is close to 0 and $t_N$ is close to 1, $\rho(t_0)$ and $\rho(t_N)$ are close to $\rho_0$ and $\rho_1$, respectively. In addition, the initialization error can be made very small (e.g., when forcing $I(t_0, x_0, x_1) = x_0$). This technique is also known as early stopping in the context of diffusion models (Song et al., 2021; Hongrui et al., 2023; Benton et al., 2024a).

**Main Ideas of the Proof** We sketch the main ideas of the proof of Theorem 4.5, with the detailed proof in Appendix D. Since we are comparing a continuous-time process with a discrete-time estimator, the first key step is to bridge this gap. We do this by reinterpreting the discrete-time process $\{\hat{X}_{t_k}\}_{k=0}^N$ as a piecewise-defined continuous process. Specifically, we define

$$\hat{X}_t = F_{t_k \to t}(\hat{X}_{t_k}) := \hat{X}_{t_k} + (t - t_k)\hat{b}(t_k, \hat{X}_{t_k})$$

for $t \in [t_k, t_{k+1}]$. Under Assumption 4.4 and for sufficiently small step sizes $h_k$, the mapping $F_{t_k \to t} : x \mapsto x + (t - t_k)\hat{b}(t_k, x)$ is a diffeomorphism on $\mathbb{R}^d$, which allows us to write $\mathrm{d}\hat{X}_t = \hat{b}(t_k, \hat{X}_{t_k})\mathrm{d}t = \hat{b}(t_k, F_{t_k \to t}^{-1}(\hat{X}_t))\mathrm{d}t$. By defining the effective drift $\tilde{b}(t, x) := \hat{b}(t_k, F_{t_k \to t}^{-1}(x))\mathrm{d}t$, we obtain $\mathrm{d}\hat{X}_t = \tilde{b}(t, \hat{X}_t)\mathrm{d}t$, which is of the same form as the ground truth ODE.

To quantify the similarity between the two processes, the main technical challenge is to show that the approximate drift $\tilde{b}(t, x)$ closely matches the true drift $b(t, x)$, as well as their respective divergences. Let $(X_t)_{t \in [0,1]}$ denote the ground truth process. We apply the following sequential approximation scheme to decompose the errors for both the drift and the divergence (see Appendix D.2 and D.3):

$$\tilde{b}(t, X_t) \approx \hat{b}(t_k, X_{t_k}) \approx b(t, X_t), \quad \text{(Drift Error)}$$

$$\nabla \cdot \tilde{b}(t, X_t) = \mathrm{tr}\left[\nabla\hat{b}(t_k, F_{t_k \to t}^{-1}(X_t)) \cdot \nabla F_{t_k \to t}(F_{t_k \to t}^{-1}(X_t))^{-1}\right]$$

$$\approx \mathrm{tr}\left[\nabla\hat{b}(t_k, X_{t_k}) \cdot I_d\right] \approx \nabla \cdot b(t, X_t). \quad \text{(Divergence Error)}$$

Combined with the analysis on partial differential equations (see Lemma D.2), we can then control the overall TV distance by the controlled drift and divergence errors.

## 4.2 Complexity of Forward Euler Method

Theorem 4.5 is built on a general stochastic interpolant framework, which is available to control the convergence rates for every valid stochastic interpolants (i.e., valid $I$ and $\gamma$) under necessary assumptions. Below, we provide a detailed analysis on the convergence complexities for some widely used stochastic interpolants.

**Stochastic Interpolants with** $\gamma(t) = \sqrt{at(1 - t)}$ For this case, $\gamma^2(t) = at(1 - t)$ is the variance of a Brownian bridge process, which is a popular choice (Albergo et al., 2023; De Bortoli et al., 2024). To optimize the schedule, we focus on minimizing the discretization error in Theorem 4.5:

$$\varepsilon \lesssim \sum_{k=0}^{N-1} h_k^2 \left[\bar{\gamma}_k^{-4} d^2 + \bar{\gamma}_k^{-2} M^2\right].$$

The error is proportional to $h_k^2 \bar{\gamma}_k^{-4}$, which suggests setting $h_k \propto \bar{\gamma}_k^2$ to balance the error. Inspired by the time-scheduling method built for SDE in Liu et al. (2025), we construct the following schedule: define a midpoint $m$ at $t_m = 0.5$ and select a step size scale parameter $h > 0$. The schedule $\{t_k\}_{k=0}^N$ is then constructed by exponentially decaying on both sides:

$$\begin{cases} t_k = \frac{1}{2}(1 - h)^{m-k}, & k \leq m; \\ t_k = 1 - \frac{1}{2}(1 - h)^{k-m}, & k > m. \end{cases} \tag{1}$$

Schedule (1) ensures $h_k \bar{\gamma}_k^{-2} = O(h)$, and the total number of steps $N = \Theta\left(\frac{1}{h}\log\left(\frac{1}{t_0(1-t_N)}\right)\right)$. By setting $\delta = \min\{t_0, 1 - t_N\}$, Theorem 4.5 directly gives the error bound for this scheduling:

$$\mathrm{TV}(\rho(t_N), \hat{\rho}(t_N)) \lesssim \mathrm{TV}(\rho(t_0), \hat{\rho}(t_0)) + \varepsilon_{\mathrm{div}} + \varepsilon_{\mathrm{drift}}\left(d^{1/2}\log^{1/2}(1/\delta) + L^{1/2}\right)$$

$$+ \underbrace{h\left[d^2\log(1/\delta) + M^2\right]}_{\text{Discretization Error}},$$

indicating that achieving an $\varepsilon$-TV error requires $N = O\left(\frac{1}{\varepsilon}\left[d^2\log^2(1/\delta) + M^2\log(1/\delta)\right]\right)$ steps.

**Variance-Preserving Diffusion Models**   Consider a stochastic interpolant defined as follows: sample $x_1 \sim \rho_{\text{data}}$ and $z \sim \mathcal{N}(0, I_d)$, then set $x_t = tx_1 + \sqrt{1-t^2}z$. Then $x_t \sim \mathcal{N}(tx_1, (1-t^2)I_d)$ when $x_1$ is given, so the marginal distributions $\rho(t)$ of the stochastic interpolant match those of the variance-preserving diffusion model (under a change of parameter, see, e.g., Albergo et al. 2023).

Similar to the previous case, we aim for $h_k \propto \gamma^2 = 1 - t^2 = \Theta(1 - t)$. This leads to the schedule:

$$t_k = 1 - (1-h)^k, \quad 0 \le k \le N, \tag{2}$$

where the scale parameter $h > 0$ controls the step sizes.

Let $\delta = 1 - t_N$ be the early-stopping time, the number of steps satisfies $N = \Theta\left(\frac{1}{h}\log\frac{1}{\delta}\right)$. If we further assume a data support bounded in each dimension, Theorem 4.5 yields that the forward Euler method with this scheduling requires $O\left(\frac{1}{\varepsilon}d^2\log^2\frac{1}{\delta}\right)$ iterations to achieve an $\varepsilon$-error, matching the prior theoretical findings (Li et al., 2024b; 2025b; 2024c). Noting that Li et al. (2024c) reports a faster $\tilde{O}(d/\varepsilon)$ complexity, their result relies on specific analysis that is restricted to Gaussian-to-data scenarios. Theorem 4.5 enables the extension to the more general data-to-data cases.

## 5   THE SECOND-ORDER RESULTS FROM THE HEUN'S METHOD

This section focuses on deriving a finite-time error bound and the corresponding iteration complexity for Heun's method, a widely used second-order approximation for solving ordinary differential equations (ODEs) (see, e.g., Stoer et al. 1980). We will then compare the error and convergence rates of Heun's method to those of the forward Euler method, highlighting the Heun's method as an acceleration of the latter.

### 5.1   ERROR BOUND FOR THE HEUN'S METHOD

Similarly, we first provide a formal definition of the Heun's method. Given a sequence of discrete time steps $\{t_k\}_{k=0}^N$, let $\hat{X}_{t_k}$ denote the estimated solution at time $t_k$. Beginning with an initial condition $\hat{X}_{t_0}$, the method proceeds via the following iterative scheme:

$$\begin{cases} \tilde{X}_{t_{k+1}} = \hat{X}_{t_k} + h_k \cdot \hat{b}(t_k, \hat{X}_{t_k}), \\ \hat{X}_{t_{k+1}} = \hat{X}_{t_k} + \frac{1}{2}h_k\left[\hat{b}(t_k, \hat{X}_{t_k}) + \hat{b}(t_{k+1}, \tilde{X}_{t_{k+1}})\right]. \end{cases} \tag{Heun}$$

Similar to the first-order case, we aim to quantify and control the error between the estimated terminal distribution and the true terminal distribution. However, to accommodate this higher-order method, we must modify some of our initial assumptions in Section 4.

**Assumption 5.1** (Regularity). *Let $\nu$ be the joint distribution of $\rho_0$ and $\rho_1$, we assume that:*

- *The **sixth** moment of the difference between the initial and goal distribution is bounded: $\mathbb{E}_{(x_0,x_1)\sim\nu}\left[\|x_0 - x_1\|^6\right] < \infty$.*

- *The time derivatives of $I(t, x_0, x_1)$ and $\gamma^2(t)$ are bounded up to the **third** order: There exist positive constants $C_I, C_\gamma > 0$ such that for all $x_0, x_1 \in \mathbb{R}^d$ and $p \in \{1, 2, 3\}$,*

$$\|\partial_t^p I(t, x_0, x_1)\| \le C_I\|x_0 - x_1\|, \quad \left|\frac{\mathrm{d}^p}{\mathrm{d}t^p}\left[\gamma^2(t)\right]\right| \le C_\gamma.$$

**Assumption 5.2** (Drift Estimation Error). *Let $\varepsilon_{1,k}(x) = \|\hat{b}(t_k, x) - b(t_k, x)\|$. Then we assume*

$$\sum_{k=0}^{N-1} h_k\mathbb{E}\left[\varepsilon_{1,k}(x_{t_k})^2 + \varepsilon_{1,k+1}(x_{t_{k+1}})^2\right] \le \varepsilon_{drift}^2 < \infty.$$

**Assumption 5.3** (Jacobian Estimation Error). *Let $\varepsilon_{2,k}(x) = \left\|\nabla\hat{b}(t_k, x) - \nabla b(t_k, x)\right\|_F$. We assume*

$$\sum_{k=0}^{N-1} h_k\mathbb{E}\left[\varepsilon_{2,k}(x_{t_k})^2 + \varepsilon_{2,k+1}(x_{t_{k+1}})^2\right]^{1/2} \le \varepsilon_{div} < \infty.$$

Assumptions 5.1, 5.2, and 5.3 are similar to Assumptions 4.1, 4.2, and 4.3, respectively, while they are adapted for the higher-order Heun's method. Assumption 5.1 requires a stricter regularity since we are analyzing higher-order integrator. Assumptions 5.2 and 5.3 incorporate additional terms, reflecting the fact that Heun's method performs two evaluations of $\hat{b}(t, x)$ per step. It is worth noting that Assumption 5.3 utilizes $\varepsilon_{2,k}^2$ instead of $\varepsilon_{2,k}$, which is slightly different with Assumption 5.3.

With the preceding assumptions established, we now present our main theorem for Heun's method.

**Theorem 5.4.** *Under Assumptions 5.1, 5.2, 5.3 and 4.4, if the equation is solved using Heun's method with an initial condition $\hat{X}_{t_0} \sim \hat{\rho}(t_0)$, and provided that the step sizes satisfy $h_k \leq \min\left\{\frac{1}{4L}, \mathbb{E}[\|x_0 - x_1\|^6]^{-1/3}, d^{-1}\bar{\gamma}_k^2\right\}$, then the TV error is bounded as:*

$$\mathrm{TV}(\rho(t_N), \hat{\rho}(t_N)) \lesssim \mathrm{TV}(\rho(t_0), \hat{\rho}(t_0)) + \varepsilon_{\mathrm{div}} + \varepsilon_{\mathrm{drift}}\left(d^{1/2} S(\gamma, t_0, t_N)^{1/2} + L^{1/2}\right)$$

$$+ \underbrace{\sum_{k=0}^{N-1} h_k^3 \left[\bar{\gamma}_k^{-6} d^3 + \bar{\gamma}_k^{-4} M^3\right]}_{\text{Discretization Error}}.$$

*Here, we have defined $\bar{\gamma}_k := \inf_{t \in [t_k, t_{k+1}]} \gamma(t)$, $S(\gamma, t_0, t_N) := \int_{t_0}^{t_N} \gamma^{-2}(t)\mathrm{d}t$, and $M := \max\left\{d, L, \mathbb{E}_\nu\left[\|x_0 - x_1\|^6\right]^{1/3}\right\}$.*

Similar to Theorem 4.5, Theorem 5.4 bounds the TV error to the target distribution. The primary distinction lies in the discretization error term, where the order of the step sizes is enhanced from $h_k^2$ to $h_k^3$, reflecting the higher-order accuracy. The $O(h_k^3)$ error for each step aligns with prior analyses on diffusion models (see, e.g., Huang et al. 2025).

**Main Ideas of the Proof**   The proof of Theorem 5.4 (see Appendix E) shares similar ideas as the first-order case. A central novelty lies in the following construction of the continuous-time interpolation for the discrete Heun updates:

$$\hat{X}_t = G_{t_k \to t}(\hat{X}_{t_k})$$
$$:= \hat{X}_{t_k} + \left[(t - t_k) - \frac{(t - t_k)^2}{2(t_{k+1} - t_k)}\right]\hat{b}(t_k, \hat{X}_{t_k}) + \frac{(t - t_k)^2}{2(t_{k+1} - t_k)}\hat{b}(t_{k+1}, F_{t_k \to t_{k+1}}(\hat{X}_{t_k})).$$

This interpolation yields a velocity $\frac{\mathrm{d}}{\mathrm{d}t}\hat{X}_t = \frac{t_{k+1}-t}{t_{k+1}-t_k}\hat{b}(t_k, \hat{X}_{t_k}) + \frac{t-t_k}{t_{k+1}-t_k}\hat{b}(t_{k+1}, F_{t_k \to t_{k+1}}(\hat{X}_{t_k}))$, which linearly interpolates between the two drift estimates and achieves higher-order accuracy. Unlike some alternatives (e.g., Huang et al. 2025), our method only requires $\hat{b}(t, x)$ at discrete timesteps.

For small $h_k$, the mapping $G_{t_k \to t}$ is similarly a diffeomorphism, enabling the representation $\frac{\mathrm{d}}{\mathrm{d}t}\hat{X}_t = \tilde{b}(t, \hat{X}_t) := \partial_t G_{t_k \to t}(G_{t_k \to t}^{-1}(\hat{X}_t))$. As in the first-order case, we establish that $\tilde{b}(t, x)$ and its divergence closely approximate their ground-truth counterparts, ultimately enabling controlling the TV error. The linear velocity interpolation yields an $O(h_k^2)$ approximation error, improving the previous $O(h_k)$ error for Euler's method. The detailed decomposition is omitted here for brevity.

## 5.2   COMPLEXITY OF HEUN'S METHOD

Similar to the forward Euler case, Theorem 5.4 can be applied to all stochastic interpolants satisfying necessary assumptions. Below, we analyze the complexity for the same stochastic interpolants discussed in Section 4.2, with the second-order Heun's method replacing the forward Euler method.

**Stochastic Interpolants with $\gamma(t) = \sqrt{at(1-t)}$**   The discretization error in Theorem 5.4 is proportional to $h_k^3 \bar{\gamma}_k^{-6}$. Hence, balancing the error suggests $h_k \propto \bar{\gamma}_k^2$ again, leading to the exponentially decaying schedule (1). Let $\delta = \min\{t_0, 1 - t_N\}$. Applying Theorem 5.4 shows that achieving an $\varepsilon$-TV error requires $N = O\left(\frac{1}{\sqrt{\varepsilon}}\left[d^{3/2}\log^{3/2}(1/\delta) + M^{3/2}\log(1/\delta)\right]\right)$ steps. This represents a reduction in complexity from $O(1/\varepsilon)$ (for forward Euler) to $O(1/\sqrt{\varepsilon})$.

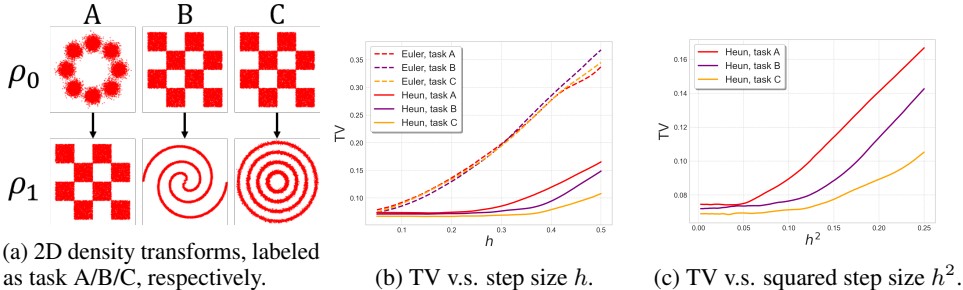

(a) 2D density transforms, labeled as task A/B/C, respectively.

(b) TV v.s. step size $h$.

(c) TV v.s. squared step size $h^2$.

Figure 1: Empirical verification of convergence rates for numerical methods.

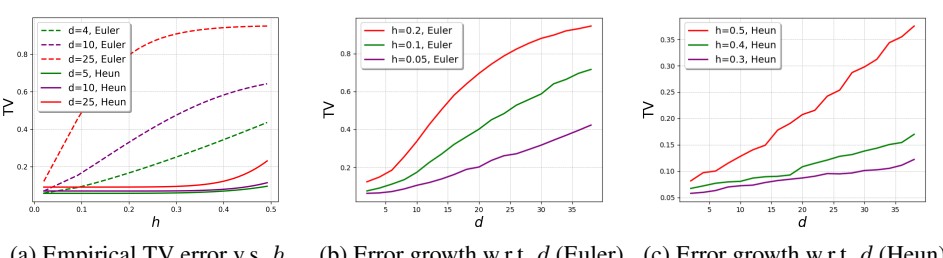

(a) Empirical TV error v.s. $h$.

(b) Error growth w.r.t. $d$ (Euler).

(c) Error growth w.r.t. $d$ (Heun).

Figure 2: Empirical TV error for $d$-dimensional Gaussian mixtures.

**Variance-Preserving Diffusion Models** Recall that the interpolant $x_t = tx_1 + \sqrt{1-t^2}z$ shares the same marginal distributions as variance-preserving diffusion models. Using our proposed schedule (2), we derive a complexity of $N = O\left(\varepsilon^{-1/2}\left[d^{3/2}\log^{3/2}(1/\delta) + M^{3/2}\log(1/\delta)\right]\right)$, where $\delta = 1 - t_N$ is the early-stopping time and $M = \max\{d, L, \mathbb{E}[\|x_1\|^6]^{1/3}\}$. When the data support is bounded in each dimension, which is a common scenario in practice, $M$ is typically $O(d)$, which simplifies the complexity to $N = O\left(\varepsilon^{-1/2}d^{3/2}\log^{3/2}(1/\delta)\right)$. This $O(\varepsilon^{-1/2})$ convergence rate is consistent with prior works (Li et al., 2024a; Huang et al., 2025). Notably, our dependence on the dimension $d$ is lower than the $\tilde{O}(\varepsilon^{-1/2}d^2)$ complexity reported by Li et al. (2025a) for a second-order method.

While Huang et al. (2025) also achieve a comparable $\tilde{O}(\varepsilon^{-1/2}d^{3/2})$ complexity, their analysis relies on stricter assumptions. Specifically, they require uniform boundedness of the third-order derivatives of the score estimator $\hat{s}(t, x)$ with respect to both $t$ and $x$, whereas we only assume bounded second-order derivatives of $\hat{b}(t, x)$ with respect to $x$. This is a significant distinction, especially since the true drift function's $t$-derivative can be unbounded. Additionally, their analysis considers errors over the entire continuous interval, while ours focuses on the discrete time steps used by Heun's method. See Appendix B for further comparison.

## 6 NUMERICAL EXPERIMENTS

This section presents experimental validation of our theories. Using the interpolant $x_t = (1-t)x_0 + tx_1 + \sqrt{2at(1-t)}z$ and schedule (1), we evaluate both forward Euler and Heun's methods on 2D datasets and $d$-dimensional Gaussian mixtures. Our experiments characterize the TV error growth w.r.t.: (i) step size scale $h$ and (ii) dimension $d$.

We evaluate our framework on three 2D dataset pairs from Grathwohl et al. (2019). We have designed three different generation tasks, as shown in Figure 1a. Using a neural network to estimate $b(t, x)$, we compare the forward Euler and Heun's methods initialized at $\rho(t_0)$. Figures 1b and 1c show the empirical TV distances between $\rho(t_N)$ and $\hat{\rho}(t_N)$, confirming our theoretical complexity analysis: the discretization error bounds are $O(h)$ (Euler) and $O(h^2)$ (Heun).

For $d$-dimensional Gaussian mixtures where $b(t, x)$ admits analytical solutions (see Albergo et al. 2023), we evaluate the empirical TV error without model training. Figure 2a demonstrates the error growth rate versus step size $h$, confirming our theoretical convergence rates. Figure 2b and 2c further examines the dimensional dependence at fixed $h$. While our theory establishes $O(hd^2)$ (Euler) and $O(h^2d^3)$ (Heun) error bounds, empirical observations suggest linear growth in both cases. This theory-experiment gap indicate that the current bound could be further improved, warranting future investigation.

In addition, to further demonstrate the applicability of our theory, we evaluate both Euler's and Heun's methods on real-world image generation tasks, as detailed in Appendix C.

## 7 CONCLUSIONS

This paper presented a finite-time analysis of discrete-time numerical implementations for ODEs derived within the stochastic interpolants framework. We established total variation distance error bounds for both the first-order forward Euler and second-order Heun's methods, quantifying the discrepancy between true and approximated target distributions. Furthermore, we analyzed the iteration complexity for both methods, elucidating their convergence rates. Numerical experiments corroborated our theoretical convergence findings.

## GRANT ACKNOWLEDGEMENT

This work is supported by the National Natural Science Foundation of China Grant 52494974.

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

## USE OF LARGE LANGUAGE MODELS

Parts of this manuscript, including sections of the main text and appendix, were polished and refined using a large language model (LLM). The LLM was exclusively used for editing and proofreading to improve clarity, grammar, and flow. All research ideas, technical content, results, and analyses were generated solely by the authors. This specific statement was also generated with the assistance of an LLM.

## NOTATIONS

Unless otherwise specified, we use the following notations throughout the paper:

- For a vector $x \in \mathbb{R}^d$, $\|x\|$ denotes its $\ell_2$ norm.
- For a matrix $A$, $\|A\|$ denotes the operator norm induced by the $\ell_2$ norm, and $\|A\|_F$ represents the Frobenius norm.
- The notations $\frac{d}{dt}$, $\frac{\partial}{\partial t}$ and $\partial_t$ all refers to the (partial) derivative with respect to $t$.
- For two vectors $x, y \in \mathbb{R}^d$, $x \cdot y$ denotes their inner product, and $x \otimes y$ denotes their tensor product.
- The notation $\nabla$ denotes the the gradient operator. For a scalar-valued function $f$, $\nabla f$ is the gradient vector. For a vector-valued function $f$, $\nabla f$ is the Jacobian matrix. Unless specified, the gradient is always taken with respect to the spatial variable $x$, not the temporal variable $t$.
- The divergence of a function $f$ is written as $\nabla \cdot f = \sum_{k=1}^d \frac{\partial f}{\partial x_k}$.
- The Laplacian of a function $f$ is denoted by $\Delta f = \sum_{k=1}^d \frac{\partial^2 f}{\partial x_k^2}$.
- We use the notation $f \lesssim g$ to indicate that $f = O(g)$.

## A   MORE DETAILS ON STOCHASTIC INTERPOLANTS

Recall that the stochastic interpolant is a stochastic process defined by

$$x_t = I(t, x_0, x_1) + \gamma(t)z.$$

Although $x_t$ establishes a connection between the two distributions, its computation depends on both $x_0$ and $x_1$, making it unsuitable as a generative model yet. To solve this problem, Albergo et al. (2023) demonstrated that the marginal density $\rho(t, x)$ satisfies the following transport equation (where the notation $\nabla$ represents the gradient operator):

$$\partial_t \rho(t, x) + \nabla \cdot (\rho(t, x)b(t, x)) = 0,$$

where the velocity field $b(t, x)$ is given by

$$b(t, x) = \mathbb{E}[\dot{x}_t | x_t = x] = \mathbb{E}[\partial_t I(t, x_0, x_1) + \dot{\gamma}_t z | x_t = x].$$

This transport equation reveals that any process $(X_t)_{t \in [0,1]}$ satisfying the initial condition $X_0 \sim \rho_0$ and solving the ODE

$$dX_t = b(t, X_t)dt,$$

will share identical marginal distributions with the stochastic interpolant $(x_t)_{t \in [0,1]}$, i.e., $X_t \sim \rho(t)$ for all $t \in [0, 1]$. Crucially, the temporal derivative of $X_t$ depends only on the current time $t$ and position $X_t$. Consequently, when initialized with a sample $x_0 \sim \rho_0$, solving $(*)$ yields $X_1 \sim \rho_1$.

Furthermore, for any non-negative function $\epsilon(t) \geq 0$, Albergo et al. (2023) shows that the solution to the SDE

$$dX_t = [b(t, X_t) + \epsilon(t)s(t, X_t)]dt + \sqrt{2\epsilon(t)}dW_t$$

also transforms samples from $\rho_0$ into $\rho_1$, where $W_t$ denotes the standard Wiener process and $s(t, x) = \nabla \log \rho(t, x) = \mathbb{E}[\gamma^{-1}z | x_t = x]$ represents the well-known score function. Notably, the ODE in $(*)$ emerges as a special case of this SDE when $\epsilon(t) \equiv 0$.

In practice, an estimator $\hat{b}(t, x)$ of the expected velocity field $b(t, x)$ can be obtained by minimizing the quadratic loss:

$$\mathcal{L}[\hat{b}] = \int_0^1 \mathbb{E}\left[\frac{1}{2}\|\hat{b}(t, x_t)\|^2 - b(t, x_t) \cdot \dot{x}_t\right] dt,$$

where $x_t \sim \rho(t)$ denotes the stochastic interpolant process. This loss differs from the mean squared error

$$\text{MSE} = \int_0^1 \mathbb{E}\left[\frac{1}{2}\|\hat{b}(t, x_t) - b(t, x_t)\|^2\right] dt$$

by a constant that is independent of $\hat{b}$, as seen by expanding the quadratic term. The resulting estimator $\hat{b}(t, x)$, when used in $(*)$, yields a generative model that transports $\rho_0$ to $\rho_1$.

Table 1: Comparison with Previous Results

| Reference | Order | Smoothness | Support | Complexity |
|---|---|---|---|---|
| Li et al. (2024b) Theorem 1 | 1 | No Requirements | Bounded | $\tilde{O}\left(\frac{d^2}{\varepsilon} + \frac{d^3}{\sqrt{\varepsilon}}\right)$ |
| Li et al. (2024c) Theorem 1 | 1 | No Requirements | Bounded | $\tilde{O}\left(\frac{d}{\varepsilon}\right)$ |
| Li et al. (2025b) Theorem 5.5 | 1 | $L$-Lipschitz | Bounded | $\tilde{O}\left(\frac{LR^4 d^2}{\varepsilon}\right)$ |
| Li et al. (2024a) Theorem 1 | 2 | No Requirements | Bounded | $\tilde{O}\left(\frac{d^3}{\sqrt{\varepsilon}}\right)$ |
| Huang et al. (2025) Theorem 3.10 | $p$ | Bounded Derivatives up to $(p+1)$-th Order[a] | Bounded | $\tilde{O}\left(\frac{(LDd)^{1+1/p}}{\varepsilon^{1/p}}\right)$ |
| Li et al. (2025a) Theorem 1 | $K$ | No Requirements | Bounded | $\tilde{O}\left(\frac{d^{1+2/K}}{\varepsilon^{1/K}}\right)$ |
| This paper Theorem 4.5 **(Data-to-Data)** | 1 | Bounded Derivatives up to 2nd Order (Assumption 4.4) | - | $O\left(\frac{M^2}{\varepsilon}\right)$ |
| This paper Theorem 4.5 **(Data-to-Data)** | 2 | Bounded Derivatives up to 2nd Order (Assumption 4.4) | - | $O\left(\frac{M^{3/2}}{\sqrt{\varepsilon}}\right)$ |

[a]They assume that $|\partial_t^k \partial_x^\alpha s_t^{(j)}(x)| \le L$ for $1 \le k + |\alpha| \le p + 1$, where $s_t$ denotes the estimated score function.

# B    COMPARISON WITH PREVIOUS WORKS

In this section, we present a comparison between our results and prior convergence analyses of ODE-based diffusion models, as summarized in Table 1. The table reports the iteration complexities of existing methods alongside their respective assumptions on the data distribution.

For Theorem 4.5 and 5.4, the value $M = \max\{d, L, \mathbb{E}_\nu[\|x_0 - x_1\|^6]^{1/3}\}$ is typically of order $O(d)$ in practical settings, such as when the data support is bounded. Under this assumption, the iteration complexity simplifies to $O\left(\frac{d^2}{\varepsilon}\right)$ and $O\left(\frac{d^{3/2}}{\sqrt{\varepsilon}}\right)$, respectively. Compared to the complexity established by Li et al. (2024a; 2025a), this bound exhibits improved dependence on the dimension $d$ when we apply second-order methods. Furthermore, relative to the result of Huang et al. (2025), when restricted to second-order solvers, our analysis relaxes the smoothness assumptions: specifically, we reduce the requirement of **third**-order smoothness in **both** $x$ and $t$ to **second**-order smoothness in $x$ **alone** (Assumption 4.4).

## B.1    MORE TECHNICAL COMPARISON WITH LI ET AL. (2025B) AND HUANG ET AL. (2025)

Our technical approach shares similarities with the works by Li et al. (2025b) and Huang et al. (2025), which also employ interpolations to reformulate discrete-time iterations into continuous-time

processes. In this section, we highlight the key technical distinctions between our analysis and prior works.

**Comparison with Li et al. (2025b)**  Although our analysis relies on Lemma 3.2 from Li et al. (2025b) (see Lemma D.2) to bound the total variation distance between the target and estimated distributions, there exist significant technical distinctions in our proofs.

First, rather than controlling the density ratio as in Li et al. (2025b) to extract the estimation error, we directly decompose the drift and divergence errors via the triangle inequality in a more intuitive manner, where estimation error and discretization error are considered in a more unified approach. Furthermore, when bounding the discretization error via time derivatives, we derive bounds in expectation rather than uniform bounds. Through a careful analysis, this leads to improved dependence on the dimension $d$ and Lipschitz constant $L$.

Second, while the general framework of Li et al. (2025b) (Theorem 4.3) can estimate distribution errors for a variety of deterministic samplers, it is primarily tailored to first-order algorithms, as their discretization error analysis only considers first-order approximations. In contrast, our error decomposition framework readily extends to the second-order Heun's method, by replacing the first-order approximation with a second-order one.

**Comparison with Huang et al. (2025)**  Huang et al. (2025) analyze general Runge–Kutta methods for probability flow ODEs in diffusion models, which include Heun's method as a special case. Compared to their results for Heun's method, our analysis offers two key improvements: (1) we reduce the smoothness requirement in $x$ from third-order to second-order, and (2) we eliminate the smoothness requirement in $t$.

For the first advantage, this advantage comes from Lemma 3.2 by Li et al. (2025b), see Lemma D.2. Huang et al. applied Gagliardo-Nirenberg inequality, which requires that a higher-order derivative is bounded, while a direct partition into drift error and divergence error has less requirements on the smoothness.

Regarding the first improvement, our approach leverages Lemma 3.2 from Li et al. (2025b) (Lemma D.2). While Huang et al. (2025) rely on the Gagliardo–Nirenberg inequality, which demands bounded higher-order derivatives, our direct decomposition into drift and divergence errors relaxes the smoothness assumptions.

As for the second improvement, we avoid explicit time-smoothness assumptions for estimator $\hat{b}(t, x)$ by implicitly utilizing the smoothness of the true drift function $b(t, x)$. This is achieved through a different interpolation scheme for the discrete-time iterates. Let $F_{t_k \to t}(x) := x + (t - t_k)\hat{b}(t_k, x)$ denote the interpolated forward Euler update. In Huang et al. (2025), Heun's method is interpolated as

$$\hat{X}_t = \hat{X}_{t_k} + \frac{t - t_k}{2}\left(\hat{b}(t_k, \hat{X}_{t_k}) + \hat{b}(t, F_{t_k \to t}(\hat{X}_{t_k}))\right),$$

whereas our analysis uses the alternative choice:

$$\hat{X}_t = \hat{X}_{t_k} + \left[(t - t_k) - \frac{(t - t_k)^2}{2(t_{k+1} - t_k)}\right]\hat{b}(t_k, \hat{X}_{t_k}) + \frac{(t - t_k)^2}{2(t_{k+1} - t_k)}\hat{b}(t_{k+1}, F_{t_k \to t_{k+1}}(\hat{X}_{t_k})).$$

Both interpolations are valid representations of Heun's method, but they differ in motivation. The interpolation in Huang et al. (2025) ensures that the Taylor expansion around $t = t_k$ closely approximates the true dynamics, requiring control over time derivatives of $\hat{b}(t, x)$. In contrast, our method approximates the intermediate drift $\frac{\mathrm{d}}{\mathrm{d}t}X_t = b(t, X_t)$ using a linear interpolation:

$$\frac{\mathrm{d}}{\mathrm{d}t}\hat{X}_t = \frac{t_{k+1} - t}{t_{k+1} - t_k}\hat{b}(t_k, \hat{X}_{t_k}) + \frac{t - t_k}{t_{k+1} - t_k}\hat{b}(t_{k+1}, \tilde{X}_{t_{k+1}}).$$

This form does not involve time derivatives of $\hat{b}(t, x)$. As long as the estimator is accurate at $t = t_k$ and $t = t_{k+1}$, our bounds only rely on the smoothness of the underlying drift $b(t, x)$, which is already established without further assumptions.

## C ADDITIONAL VERIFICATION ON GAUSSIAN-TO-IMAGE GENERATION

This section presents additional experiments on image generation from Gaussian noise, employing the same definition of the stochastic interpolant introduced in the main text.

Recall that in VP diffusion models, the noisy data at step $s$ is given by

$$x_s = \sqrt{\bar{\alpha}_s} x_{\text{data}} + \sqrt{1 - \bar{\alpha}_s} z,$$

where $z \sim \mathcal{N}(0, I_d)$ is the Gaussian noise. To align this with the stochastic interpolant framework, we define the time variable as $t = \sqrt{\bar{\alpha}_s}$ and construct the stochastic interpolant as

$$x_t = t x_1 + \sqrt{1 - t^2} z$$

with $x_1 \sim \rho_{\text{data}}$ and $z \sim \mathcal{N}(0, I_d)$. The corresponding drift term can then be expressed as

$$b(t, x) = \mathbb{E}[\dot{x}_t | x_t = x] = -\frac{t}{\sqrt{1 - t^2}} \mathbb{E}[z | x_t] + \mathbb{E}[x_1 | x_t] = \frac{x}{t} + \left( t + \frac{1 - t^2}{t} \right) s(t, x).$$

In our experiments, we leverage pre-trained score functions from Huggingface (von Platen et al., 2022) and compare the performance of the forward Euler method and Heun's method using the drift $b(t, x)$ defined above. The resulting generated images are displayed in Figure 3 and Figure 4. As illustrated, although both methods exhibit converging errors as the number of steps increases, Heun's second-order method achieves faster convergence, consistent with our theoretical predictions.

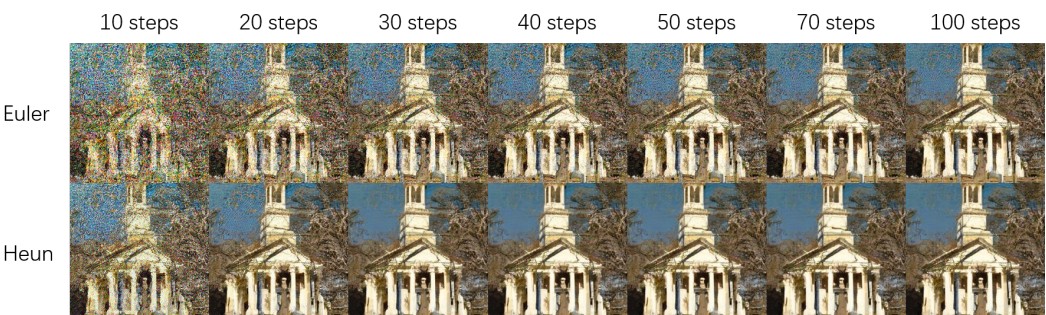

Figure 3: Comparison of Euler's and Heun's method on LSUN-Churches dataset.

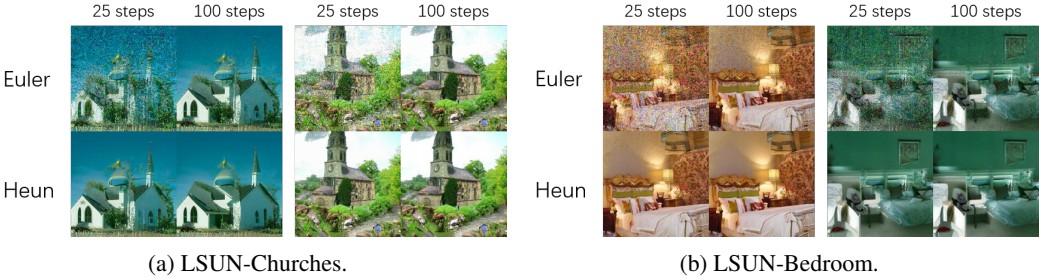

(a) LSUN-Churches.             (b) LSUN-Bedroom.

Figure 4: More comparison between Euler's and Heun's method on image datasets.

## D PROOF OF THEOREM 4.5

### D.1 INTERPOLATION OF THE DISCRETE-TIME PROCESS

First, for the forward Euler solver, we apply the following interpolation to reformulate the process as a continuous-time process:

$$\hat{X}_t = F_{t_k \to t}(\hat{X}_{t_k}) := \hat{X}_{t_k} + (t - t_k)\hat{b}(t_k, \hat{X}_{t_k}), \quad \forall t \in [t_k, t_{k+1}).$$

That is,

$$\mathrm{d}\hat{X}_t = \hat{b}(t_k, \hat{X}_{t_k})\mathrm{d}t = \partial_t F_{t_k \to t}(\hat{X}_{t_k})\mathrm{d}t.$$

To write $\hat{b}(t_k, \hat{X}_{t_k})$ as a function of $(t, \hat{X}_t)$, we need to show that $F_{t_k \to t}$ is a diffeomorphism from $\mathbb{R}^d$ to itself.

**Lemma D.1.** *Under Assumption 4.4, suppose that the step size $h_k \leq \frac{1}{2L}$, then $F_{t_k \to t}$ is a diffeomorphism, and*

$$\forall x \in \mathbb{R}^d, \quad \|\nabla F_{t_k \to t}(x)\| \leq 2, \quad \|\nabla F_{t_k \to t}^{-1}(x)\| \leq 2.$$

*Proof of Lemma D.1.*

$$\nabla F_{t_k \to t}(x) = I_d + (t - t_k)\nabla \hat{b}(t_k, x).$$

So,

$$\|\nabla F_{t_k \to t}(x) - I_d\| = (t - t_k)\|\nabla \hat{b}(t_k, x)\| \leq h_k L \leq \frac{1}{2} < 1.$$

The above inequality shows that the Jacobi matrix $\nabla F_{t_k \to t}(x)$ is invertible, so $F_{t_k \to t}$ is a local diffeomprhism. At the same time, note that $\|F_{t_k \to t}(x) - F_{t_k \to t}(y)\| \geq \frac{1}{2}\|x - y\|$, so by Hadamard's global inverse function theorem, it is a global diffeomorphism on $\mathbb{R}^d$. Moreover, the matrix norm of the inverse of its Jacobi matrix satisfies $\|\nabla F_{t_k \to t}(x)^{-1}\| \leq 2$. $\qquad\square$

By Lemma D.1, we know that $\hat{X}_t$ satisfies the ODE

$$\mathrm{d}\hat{X}_t = \tilde{b}(t, \hat{X}_t)\mathrm{d}t = \hat{b}(t_k, F_{t_k \to t}^{-1}(\hat{X}_t))\mathrm{d}t.$$

Now, we introduce an existing result by Li et al. (2025b) to control the overall TV distance.

**Lemma D.2.** *(Lemma 3.2 by Li et al. 2025b) For two processes $X_t$ and $\hat{X}_t$ solving $\mathrm{d}X_t = b(t, X_t)\mathrm{d}t$ and $\mathrm{d}\hat{X}_t = \hat{b}(t, \hat{X}_t)\mathrm{d}t$, respectively. Let $\rho(t)$ and $\hat{\rho}(t)$ be their corresponding laws. Then:*

$$\frac{\mathrm{d}}{\mathrm{d}t}\mathrm{TV}(\rho(t), \hat{\rho}(t)) = \int_{\Omega_t} (\nabla \cdot b(t, x) - \nabla \cdot \hat{b}(t, x))\rho(t, x)\mathrm{d}x$$

$$- \int_{\Omega_t} (b(t, x) - \hat{b}(t, x))\nabla \log \rho(t, x)\rho(t, x)\mathrm{d}x,$$

*where $\Omega_t = \{x \in \mathbb{R}^d : \hat{\rho}(t, x) > \rho(t, x)\}$.*

So according to Lemma D.2,

$$\mathrm{TV}(\rho(t_N), \hat{\rho}(t_N)) \leq \mathrm{TV}(\rho(t_0), \hat{\rho}(t_0))$$

$$+ \int_{t_0}^{t_N} \mathbb{E}_{X_t \sim \rho(t)} \left[ |\nabla \cdot \tilde{b}(t, X_t) - \nabla \cdot b(t, X_t)| \right] \mathrm{d}t$$

$$+ \int_{t_0}^{t_N} \mathbb{E}_{X_t \sim \rho(t)} \left[ \|\tilde{b}(t, X_t) - b(t, X_t)\| \cdot \|\nabla \ln \rho(t, X_t)\| \right] \mathrm{d}t.$$

In the following sections, we denote by $(X_t)_{t \in [t_0, t_N]}$ the true solution of the original ODE. When $t \in [t_k, t_{k+1})$ are given, we denote $z = F_{t_k \to t}^{-1}(X_t)$ for simplicity. The rest of the problem is to control the drift error $\tilde{b}(t, X_t) - b(t, X_t)$ and the divergence error $\nabla \cdot \tilde{b}(t, X_t) - \nabla \cdot b(t, X_t)$.

### D.2 CONTROLLING THE DRIFT ERROR

By a simple triangle inequality,

$$\|\tilde{b}(t, X_t) - b(t, X_t)\| \leq \underbrace{\|\hat{b}(t_k, z) - \hat{b}(t_k, X_{t_k})\|}_{A}$$

$$+ \underbrace{\|\hat{b}(t_k, X_{t_k}) - b(t_k, X_{t_k})\|}_{B}$$

$$+ \underbrace{\|b(t_k, X_{t_k}) - b(t, X_t)\|}_{C}.$$

Below we discuss the above terms respectively. For simplicity, we use the notation $\varepsilon_{1,k}(x) = \|\hat{b}(t_k, x) - b(t_k, x)\|$ to denote the error of $\hat{b}(t, x)$ at $(t_k, x)$.

For the term $A$, by Assumption 4.4,

$$
\begin{aligned}
\|\hat{b}(t_k, z) - \hat{b}(t_k, X_{t_k})\| &\leq L\|z - X_{t_k}\| \\
&= L\|F_{t_k \to t}^{-1}(X_t) - F_{t_k \to t}^{-1}(F_{t_k \to t}(X_{t_k}))\| \\
&\leq 2L\|X_t - F_{t_k \to t}(X_{t_k})\|,
\end{aligned}
$$

where the last inequality uses Lemma D.1 and the assumption that $h_k \leq \frac{1}{2L}$. For the term $C$, we introduce the following lemma:

**Lemma D.3.**

$$
\|X_t - F_{t_k \to t}(X_{t_k})\| \leq (t - t_k) \int_{t_k}^t \left\| \frac{\mathrm{d}}{\mathrm{d}s} (b(s, X_s)) \right\| \mathrm{d}s + (t - t_k)\varepsilon_{1,k}(X_{t_k})
$$

*Proof of Lemma D.3.* Consider taking derivatives and then integrate, we obtain that

$$
\begin{aligned}
\|X_t - F_{t_k \to t}(X_{t_k})\| &= \left\| \int_{t_k}^t \left( \frac{\mathrm{d}}{\mathrm{d}s} X_s - \frac{\mathrm{d}}{\mathrm{d}s} F_{t_k \to s}(X_{t_k}) \right) \mathrm{d}s \right\| \\
&= \left\| \int_{t_k}^t \left( b(s, X_s) - \hat{b}(t_k, X_{t_k}) \right) \mathrm{d}s \right\| \\
&\leq \int_{t_k}^t \|b(s, X_s) - b(t_k, X_{t_k})\| \mathrm{d}s + (t - t_k)\varepsilon_{1,k}(X_{t_k}) \\
&\leq \int_{t_k}^t \int_{t_k}^s \left\| \frac{\mathrm{d}}{\mathrm{d}u} (b(u, X_u)) \right\| \mathrm{d}u + (t - t_k)\varepsilon_{1,k}(X_{t_k}) \\
&\leq (t - t_k) \int_{t_k}^t \left\| \frac{\mathrm{d}}{\mathrm{d}s} (b(s, X_s)) \right\| \mathrm{d}s + (t - t_k)\varepsilon_{1,k}(X_{t_k}).
\end{aligned}
$$

$\square$

Then, by Lemma D.3, term $A$ can be bounded below:

$$
\begin{aligned}
\|\hat{b}(t_k, z) - \hat{b}(t_k, X_{t_k})\| &\leq 2L(t - t_k) \int_{t_k}^t \left\| \frac{\mathrm{d}}{\mathrm{d}s} (b(s, X_s)) \right\| \mathrm{d}s + \varepsilon_{1,k}(X_{t_k}) \\
&\leq \int_{t_k}^t \left\| \frac{\mathrm{d}}{\mathrm{d}s} (b(s, X_s)) \right\| \mathrm{d}s + \varepsilon_{1,k}(X_{t_k}),
\end{aligned}
$$

where the last inequality uses $h_k \leq \frac{1}{2L}$.

For the term $B$, it is just $\varepsilon_{1,k}(X_{t_k})$.

For the term $C$, apply the similar method as term $A$:

$$
\|b(t_k, X_{t_k}) - b(t, X_t)\| \leq \int_{t_k}^t \left\| \frac{\mathrm{d}}{\mathrm{d}s} b(s, X_s) \right\| \mathrm{d}s.
$$

Hence, by summing up the above results, we obtain

$$\mathbb{E}_{X_t \sim \rho(t)} \left[ \|\tilde{b}(t, X_t) - b(t, X_t)\| \cdot \|\nabla \ln \rho(t, X_t)\| \right]$$

$$\stackrel{(a)}{\leq} \mathbb{E}_{X_t \sim \rho(t)} \left[ \|\tilde{b}(t, X_t) - b(t, X_t)\|^{4/3} \right]^{3/4} \cdot \mathbb{E}_{X_t \sim \rho(t)} \left[ \|s(t, X_t)\|^4 \right]^{1/4}$$

$$\stackrel{(b)}{\lesssim} \gamma(t)^{-1} d^{1/2} \mathbb{E}_{X_t \sim \rho(t)} \left[ \left( \int_{t_k}^t \left\| \frac{\mathrm{d}}{\mathrm{d}s} b(s, X_s) \right\| \mathrm{d}s + \varepsilon_{1,k}(X_{t_k}) \right)^{4/3} \right]^{3/4}$$

$$\stackrel{(c)}{\lesssim} \gamma(t)^{-1} d^{1/2} \mathbb{E}_{X_t \sim \rho(t)} \left[ (t - t_k)^{1/3} \int_{t_k}^t \left\| \frac{\mathrm{d}}{\mathrm{d}s} b(s, X_s) \right\|^{4/3} \mathrm{d}s + \varepsilon_{1,k}(X_{t_k})^{4/3} \right]^{3/4}$$

$$\lesssim \gamma(t)^{-1} d^{1/2} \left[ (t - t_k)^{1/3} \int_{t_k}^t \mathbb{E} \left[ \left\| \frac{\mathrm{d}}{\mathrm{d}s} b(s, X_s) \right\|^{4/3} \right] + \mathbb{E} \left[ \varepsilon_{1,k}(X_{t_k})^{4/3} \right] \right]^{3/4}$$

$$\stackrel{(d)}{\lesssim} \gamma(t)^{-1} d^{1/2} \mathbb{E} \left[ \varepsilon_{1,k}(X_{t_k})^2 \right]^{1/2}$$

$$+ \gamma(t)^{-1} d^{1/2} (t - t_k) \left[ \mathbb{E} \left[ \|x_0 - x_1\|^4 \right]^{1/2} \bar{\gamma}_k^{-1} d^{1/2} + \bar{\gamma}_k^{-3} d^{3/2} \right]$$

$$\lesssim \gamma(t)^{-1} d^{1/2} \mathbb{E} \left[ \varepsilon_{1,k}(X_{t_k})^2 \right]^{1/2} + h_k \bar{\gamma}_k^{-2} d \mathbb{E} \left[ \|x_0 - x_1\|^4 \right]^{1/2} + h_k \bar{\gamma}_k^{-4} d^2.$$

Here, (a) applies Hölder's inequality with power $4$ and $4/3$; (b) plugs in the results for terms $A$, $B$ and $C$; (c) applies the Jensen's inequality to the integral; and (d) uses Lemma F.13 to obtain the bound.

Therefore,

$$\int_{t_0}^{t_N} \mathbb{E}_{X_t \sim \rho(t)} \left[ \|\tilde{b}(t, X_t) - b(t, X_t)\| \cdot \|\nabla \ln \rho(t, X_t)\| \right] \mathrm{d}t$$

$$\lesssim d^{1/2} \int_{t_0}^{t_N} \gamma(t)^{-1} \mathbb{E} \left[ \varepsilon_{1,k}(X_{t_k})^2 \right]^{1/2} \mathrm{d}t$$

$$+ \sum_{k=0}^{N-1} h_k^2 \left[ \bar{\gamma}_k^{-2} d \mathbb{E} \left[ \|x_0 - x_1\|^4 \right]^{1/2} + \bar{\gamma}_k^{-4} d^2 \right]$$

$$\stackrel{(a)}{\lesssim} d^{1/2} \sqrt{\int_{t_0}^{t_N} \gamma(t)^{-2} \mathrm{d}t} \sqrt{\int_{t_0}^{t_N} \mathbb{E} \left[ \varepsilon_{1,k}(X_{t_k})^2 \right]^{1/2} \mathrm{d}t}$$

$$+ \sum_{k=0}^{N-1} h_k^2 \left[ \bar{\gamma}_k^{-2} d \mathbb{E} \left[ \|x_0 - x_1\|^4 \right]^{1/2} + \bar{\gamma}_k^{-4} d^2 \right]$$

$$\stackrel{(b)}{\leq} d^{1/2} \varepsilon_{\mathrm{drift}} S(\gamma, t_0, t_N)^{1/2} + \sum_{k=0}^{N-1} h_k^2 \left[ \bar{\gamma}_k^{-2} d \mathbb{E} \left[ \|x_0 - x_1\|^4 \right]^{1/2} + \bar{\gamma}_k^{-4} d^2 \right],$$

where

$$S(\gamma, t_0, t_N) = \int_{t_0}^{t_N} \gamma(t)^{-2} \mathrm{d}t. \tag{3}$$

For the above derivation, the inequality (a) applies Cauchy-Schwarz inequality, while the inequality (b) uses Assumption 4.2.

### D.3 CONTROLLING THE DIVERGENCE ERROR

First, according to the chain rule of derivatives,

$$\nabla \tilde{b}(t, X_t) = \nabla \hat{b}(t_k, z) \cdot \nabla F_{t_k \to t}(z)^{-1}.$$

Therefore, by applying triangle inequality,

$$\left|\nabla \cdot \tilde{b}(t, X_t) - \nabla \cdot b(t, X_t)\right| \le \underbrace{\left|\text{tr}\left[\left[\nabla\hat{b}(t_k, z) - \nabla\hat{b}(t_k, X_{t_k})\right] \cdot \nabla F_{t_k \to t}(z)^{-1}\right]\right|}_{A}$$

$$+ \underbrace{\left|\text{tr}\left[\left[\nabla\hat{b}(t_k, X_{t_k}) - \nabla b(t_k, X_{t_k})\right] \cdot \nabla F_{t_k \to t}(z)^{-1}\right]\right|}_{B}$$

$$+ \underbrace{\left|\text{tr}\left[\nabla b(t_k, X_{t_k}) \cdot \left(\nabla F_{t_k \to t}(z)^{-1} - I_d\right)\right]\right|}_{C}$$

$$+ \underbrace{\left|\nabla \cdot b(t_k, X_{t_k}) - \nabla \cdot b(t, X_t)\right|}_{D}.$$

Now, we deal with the above terms respectively. Similarly to the velocity error, we use the notation $\varepsilon_{2,k}(x) = \|\nabla\hat{b}(t_k, x) - \nabla b(t_k, x)\|_F$.

First, since $\|\nabla F_{t_k \to t}(z) - I_d\|_F \le Lh_k \le \frac{1}{2}$,

$$\|\nabla F_{t_k \to t}(z)^{-1} - I_d\|_F = \|(I_d + F_{t_k \to t}(z) - I_d)^{-1} - I_d\|_F$$

$$\le \|I_d + \sum_{i=1}^{\infty}(I_d - F_{t_k \to t}(z))^i - I_d\|_F$$

$$\le \sum_{i=1}^{\infty}(Lh_k)^i \le 2Lh_k.$$

So for the term $A$,

$$\left|\text{tr}\left[\left[\nabla\hat{b}(t_k, z) - \nabla\hat{b}(t_k, X_{t_k})\right] \cdot \nabla F_{t_k \to t}(z)^{-1}\right]\right|$$

$$\le |\nabla \cdot \hat{b}(t_k, z) - \nabla \cdot \hat{b}(t_k, X_{t_k})| + 2Lh_k\|\nabla\hat{b}(t_k, z) - \nabla\hat{b}(t_k, X_{t_k})\|_F$$

$$\le 2L^{3/2}\|z - X_{t_k}\| \le 4L^{3/2}\|X_t - F_{t_k \to t}(X_{t_k})\|$$

$$\le 2L^{1/2}\int_{t_k}^{t}\left\|\frac{\text{d}}{\text{d}s}(b(s, X_s))\right\|\text{d}s + 2L^{1/2}\varepsilon_{1,k}(X_{t_k}),$$

where the last inequality applies Lemma D.3.

For the term $B$,

$$\left|\text{tr}\left[\left[\nabla\hat{b}(t_k, X_{t_k}) - \nabla b(t_k, X_{t_k})\right] \cdot \nabla F_{t_k \to t}(z)^{-1}\right]\right|$$

$$\le 2\left\|\nabla\hat{b}(t_k, X_{t_k}) - \nabla b(t_k, X_{t_k})\right\|_F$$

$$\le 2\varepsilon_{2,k}(X_{t_k}).$$

For the term $C$,

$$\left|\text{tr}\left[\nabla b(t_k, X_{t_k}) \cdot \left(\nabla F_{t_k \to t}(z)^{-1} - I_d\right)\right]\right| \le 2Lh_k\|\nabla b(t_k, X_{t_k})\|_F.$$

For the term $D$, consider taking derivatives w.r.t. $t$:

$$|\nabla \cdot b(t_k, X_{t_k}) - \nabla \cdot b(t, X_t)| \le \int_{t_k}^{t}\left|\frac{\text{d}}{\text{d}s}(\nabla \cdot b(s, X_s))\right|\text{d}s.$$

Therefore, combining Lemmas F.14, F.13 and F.1, we can get

$$\mathbb{E}\left[\left|\nabla \cdot \tilde{b}(t, X_t) - \nabla \cdot b(t, X_t)\right|\right]$$

$$\lesssim h_k\left[\mathbb{E}\left[\|x_0 - x_1\|^3\right]^{2/3}(\bar{\gamma}_k^{-2}d + \bar{\gamma}_k^{-1}d^{1/2}L^{1/2}) + \bar{\gamma}_k^{-4}d^2\right.$$

$$+ L\bar{\gamma}_k^{-1}d^{1/2}\sqrt{\mathbb{E}\|x_0 - x_1\|^2} + L\bar{\gamma}_k^{-2}d\big]$$

$$+ L^{1/2}\mathbb{E}\left[\varepsilon_{1,k}(X_{t_k})\right] + \mathbb{E}\left[\varepsilon_{2,k}(X_{t_k})\right].$$

Then, taking the integral and applying Assumption 4.2 and 4.3:

$$\int_{t_0}^{t_N} \mathbb{E}_{X_t \sim \rho(t)} \left[ |\nabla \cdot \tilde{b}(t, X_t) - \nabla \cdot b(t, X_t)| \right] \mathrm{d}t$$

$$\lesssim \sum_{k=0}^{N-1} h_k^2 \left[ \mathbb{E}\left[\|x_0 - x_1\|^3\right]^{2/3} \bar{\gamma}_k^{-2} d + \bar{\gamma}_k^{-4} d^2 \right.$$

$$\left. + L\bar{\gamma}_k^{-1} d^{1/2} \sqrt{\mathbb{E}\|x_0 - x_1\|^2} + L\bar{\gamma}_k^{-2} d \right]$$

$$+ \int_{t_0}^{t_N} \mathbb{E}\left[\varepsilon_{2,k}(X_{t_k})\right] \mathrm{d}t + \int_{t_0}^{t_N} \mathbb{E}\left[\varepsilon_{1,k}(X_{t_k})\right] \mathrm{d}t$$

$$\lesssim \varepsilon_{\mathrm{div}} + L^{1/2}\varepsilon_{\mathrm{drift}} + \sum_{k=0}^{N-1} h_k^2 \left[ \mathbb{E}\left[\|x_0 - x_1\|^3\right]^{2/3} \bar{\gamma}_k^{-2} d + \bar{\gamma}_k^{-4} d^2 \right.$$

$$\left. + L\bar{\gamma}_k^{-1} d^{1/2} \sqrt{\mathbb{E}\|x_0 - x_1\|^2} + L\bar{\gamma}_k^{-2} d \right].$$

Combining Lemma D.2, the bound on velocity error and the bound on divergence error, we can obtain the error bound given in Theorem 4.5.

## E  PROOF OF THEOREM 5.4

### E.1  INTERPOLATION OF THE DISCRETE-TIME PROCESS

The proof of Theorem 5.4 shares the same basic ideas as Theorem 4.5. In order to compare the discrete-time iterations with the true process, we first apply an interpolation on the Heun's method: $\hat{X}_t = G_{t_k \to t}(\hat{X}_{t_k})$. The interpolation is defined as follows:

$$G_{t_k \to t}(x) := x + \int_{t_k}^t \left[ \hat{b}(t_k, x) + \frac{s - t_k}{t_{k+1} - t_k} \left( \hat{b}(t_{k+1}, F_{t_k \to t_{k+1}}(x)) - \hat{b}(t_k, x) \right) \right] \mathrm{d}s$$

$$= x + \left[ (t - t_k) - \frac{(t - t_k)^2}{2(t_{k+1} - t_k)} \right] \hat{b}(t_k, x) + \frac{(t - t_k)^2}{2(t_{k+1} - t_k)} \hat{b}(t_{k+1}, F_{t_k \to t_{k+1}}(x)),$$

where recall that $F_{t_k \to t}(x) := x + (t - t_k)\hat{b}(t_k, x)$ is the interpolation for the Euler's method.

Similarly, we want to show that $\hat{X}_t$ admits the same ODE form as the true process, which relies on the following Lemma.

**Lemma E.1.** *Suppose Assumption 4.4 holds, then if $h_k \leq \frac{1}{4L}$, $G_{t_k \to t}$ is a diffeomprhism, and for all $x \in \mathbb{R}^d$,*

$$\|\nabla G_{t_k \to t}(x)\| \leq 2, \quad \|\nabla G_{t_k \to t}(x)^{-1}\| \leq 2.$$

*Proof.* Similarly to the proof of Lemma D.1, we only need to show that $\|\nabla G_{t_k \to t}(x) - I_d\| \leq \frac{1}{2}$.

$$\nabla G_{t_k \to t}(x) = I_d + \left[ (t - t_k) - \frac{(t - t_k)^2}{2(t_{k+1} - t_k)} \right] \nabla \hat{b}(t_k, x)$$

$$+ \frac{(t - t_k)^2}{2(t_{k+1} - t_k)} \nabla \hat{b}(t_{k+1}, F_{t_k \to t_{k+1}}(x)) \nabla F_{t_k \to t_{k+1}}(x)$$

$$= I_d + \left[ (t - t_k) - \frac{(t - t_k)^2}{2(t_{k+1} - t_k)} \right] \nabla \hat{b}(t_k, x)$$

$$+ \frac{(t - t_k)^2}{2(t_{k+1} - t_k)} \nabla \hat{b}(t_{k+1}, F_{t_k \to t_{k+1}}(x)) \left( I_d + h_k \nabla \hat{b}(t_k, x) \right).$$

So,

$$\|\nabla G_{t_k \to t}(x) - I_d\| \le h_k L + \frac{1}{2} h_k L \left(1 + h_k L\right) < \frac{1}{2},$$

which completes the proof. $\qquad\square$

By Lemma E.1, if we write $z = G_{t_k \to t}^{-1}(X_t)$, then

$$\frac{\mathrm{d}}{\mathrm{d}t} \hat{X}_t = \tilde{b}(t, \hat{X}_t) := \frac{t_{k+1} - t}{t_{k+1} - t_k} \hat{b}(t_k, z) + \frac{t - t_k}{t_{k+1} - t_k} \hat{b}(t_{k+1}, F_{t_k \to t_{k+1}}(z)).$$

Then by Lemma D.2,

$$\begin{aligned}
\mathrm{TV}(\rho(t_N), \hat{\rho}(t_N)) &\le \mathrm{TV}(\rho(t_0), \hat{\rho}(t_0)) \\
&+ \int_{t_0}^{t_N} \mathbb{E}_{X_t \sim \rho(t)} \left[ |\nabla \cdot \tilde{b}(t, X_t) - \nabla \cdot b(t, X_t)| \right] \mathrm{d}t \\
&+ \int_{t_0}^{t_N} \mathbb{E}_{X_t \sim \rho(t)} \left[ \|\tilde{b}(t, X_t) - b(t, X_t)\| \cdot \|\nabla \ln \rho(t, X_t)\| \right] \mathrm{d}t.
\end{aligned}$$

## E.2 CONTROLLING THE DRIFT ERROR

Still, use the notation $\varepsilon_{1,k}(x) = \|\hat{b}(t_k, x) - b(t_k, x)\|$, then we can control $\tilde{b}(t, x) - b(t, x)$ in the following way:

$$\begin{aligned}
&\tilde{b}(t, X_t) - b(t, X_t) \\
&= \hat{b}(t_k, z) + \frac{t - t_k}{t_{k+1} - t_k} \left( \hat{b}(t_{k+1}, F_{t_k \to t_{k+1}}(z)) - \hat{b}(t_k, z) \right) - b(t, X_t) \\
&= \underbrace{\frac{t_{k+1} - t}{t_{k+1} - t_k} \left[ \hat{b}(t_k, z) - \hat{b}(t_k, X_{t_k}) \right] + \frac{t - t_k}{t_{k+1} - t_k} \left[ \hat{b}(t_{k+1}, F_{t_k \to t_{k+1}}(z)) - \hat{b}(t_{k+1}, X_{t_{k+1}}) \right]}_{A: \text{bias error}} \\
&\quad + \underbrace{\frac{t_{k+1} - t}{t_{k+1} - t_k} \left[ \hat{b}(t_k, X_{t_k}) - b(t_k, X_{t_k}) \right] + \frac{t - t_k}{t_{k+1} - t_k} \left[ \hat{b}(t_{k+1}, X_{t_{k+1}}) - b(t_{k+1}, X_{t_{k+1}}) \right]}_{B: \text{estimation error}} \\
&\quad + \underbrace{b(t_k, X_{t_k}) + \frac{t - t_k}{t_{k+1} - t_k} \left( b(t_{k+1}, X_{t_{k+1}}) - b(t_k, X_{t_k}) \right) - b(t, X_t)}_{C: \text{discretization error}}.
\end{aligned}$$

Below we bound the terms respectively. First, for the bias error $A$,

$$\begin{aligned}
A &\le \frac{t_{k+1} - t}{t_{k+1} - t_k} L \|z - X_{t_k}\| + \frac{t - t_k}{t_{k+1} - t_k} L \|F_{t_k \to t_{k+1}}(z) - X_{t_{k+1}}\| \\
&\le \frac{t_{k+1} - t}{t_{k+1} - t_k} L \|z - X_{t_k}\| \\
&\quad + \frac{t - t_k}{t_{k+1} - t_k} L \left( \|F_{t_k \to t_{k+1}}(z) - F_{t_k \to t_{k+1}}(X_{t_k})\| + \|F_{t_k \to t_{k+1}}(X_{t_k}) - X_{t_{k+1}}\| \right) \\
&\le 2L \|z - X_{t_k}\| + L(t - t_k) \int_{t_k}^{t_{k+1}} \left\| \frac{\mathrm{d}}{\mathrm{d}s} b(s, X_s) \right\| \mathrm{d}s + L(t - t_k) \varepsilon_{1,k}(X_{t_k}).
\end{aligned}$$

The last step uses Lemma D.3.

For $\|z - X_{t_k}\|$, we can observe that

$$\|z - X_{t_k}\| \le \|G_{t_k \to t}^{-1}(X_t) - G_{t_k \to t}^{-1}(G_{t_k \to t}(X_{t_k}))\| \le 2\|X_t - G_{t_k \to t}(X_{t_k})\|.$$

Now, we introduce the following lemma:

**Lemma E.2.** *When $h_k \leq \frac{1}{L}$, suppose Assumption 4.4 holds,*

$$\|X_t - G_{t_k \to t}(X_{t_k})\| \lesssim (t - t_k)\varepsilon_{1,k}(X_{t_k}) + (t - t_k)\varepsilon_{1,k+1}(X_{t_{k+1}})$$
$$+ (t - t_k)^2 L \int_{t_k}^{t_{k+1}} \left\| \frac{\mathrm{d}}{\mathrm{d}s} b(s, X_s) \right\| \mathrm{d}s$$
$$+ (t - t_k)(t_{k+1} - t_k) \int_{t_k}^{t_{k+1}} \left\| \frac{\mathrm{d}^2}{\mathrm{d}s^2} b(s, X_s) \right\| \mathrm{d}s.$$

*Proof of Lemma E.2.* First partition the error into several parts.

$$\|F_t(X_{t_k}) - X_t\|$$
$$\leq \|F_t(X_{t_k}) - X_{t_k} - (X_t - X_{t_k})\|$$
$$\leq \underbrace{\left[ (t - t_k) - \frac{(t - t_k)^2}{2(t_{k+1} - t_k)} \right] \cdot \left\| \hat{b}(t_k, X_{t_k}) - b(t_k, X_{t_k}) \right\|}_{\text{(i)}}$$
$$+ \underbrace{\frac{(t - t_k)^2}{2(t_{k+1} - t_k)} \cdot \left\| \hat{b}(t_{k+1}, X_{t_{k+1}}) - b(t_{k+1}, X_{t_{k+1}}) \right\|}_{\text{(ii)}}$$
$$+ \underbrace{\frac{(t - t_k)^2}{2(t_{k+1} - t_k)} \cdot \left\| \hat{b}(t_{k+1}, F_{t_k \to t_{k+1}}(X_{t_k})) - \hat{b}(t_{k+1}, X_{t_{k+1}}) \right\|}_{\text{(iii)}}$$
$$+ \underbrace{\left\| \int_{t_k}^{t} \left[ b(t_k, X_{t_k}) + \frac{s - t_k}{t_{k+1} - t_k}(b(t_{k+1}, X_{t_{k+1}}) - b(t_k, X_{t_k})) - b(s, X_s) \right] \mathrm{d}s \right\|}_{\text{(iv)}}.$$

We then need to provide upper bounds for the parts. First,

$$\text{(i)} + \text{(ii)} \leq 2(t - t_k)\varepsilon_{1,k}(X_{t_k}) + (t - t_k)\varepsilon_{1,k+1}(X_{t_{k+1}}).$$

For the third term, apply Lemma D.3,

$$\text{(iii)} \leq \frac{(t - t_k)^2}{(t_{k+1} - t_k)} L \|F_{t_k \to t_{k+1}}(X_{t_k}) - X_{t_{k+1}}\|$$
$$\leq (t - t_k)^2 L \int_{t_k}^{t_{k+1}} \left\| \frac{\mathrm{d}}{\mathrm{d}s} b(s, X_s) \right\| \mathrm{d}s + (t - t_k)^2 L \varepsilon_{1,k}(X_{t_k}).$$

For the last term, note that

$$\frac{t - t_k}{t_{k+1} - t_k} \left[ b(t_{k+1}, X_{t_{k+1}}) - b(t_k, X_{t_k}) \right] - [b(t, X_t) - b(t_k, X_{t_k})]$$
$$= \frac{t - t_k}{t_{k+1} - t_k} \int_{t_k}^{t_{k+1}} \frac{\mathrm{d}}{\mathrm{d}s} b(s, X_s) \mathrm{d}s - \int_{t_k}^{t} \frac{\mathrm{d}}{\mathrm{d}s} b(s, X_s) \mathrm{d}s$$
$$= \frac{t - t_k}{t_{k+1} - t_k} \int_{t_k}^{t_{k+1}} \left[ \frac{\mathrm{d}}{\mathrm{d}s} b(s, X_s) - \frac{\mathrm{d}}{\mathrm{d}t} b(t, X_t) \right] \mathrm{d}s - \int_{t_k}^{t} \left[ \frac{\mathrm{d}}{\mathrm{d}s} b(s, X_s) - \frac{\mathrm{d}}{\mathrm{d}t} b(t, X_t) \right] \mathrm{d}s$$
$$= \frac{t - t_k}{t_{k+1} - t_k} \int_{t}^{t_{k+1}} \mathrm{d}s \int_{t}^{s} \frac{\mathrm{d}^2}{\mathrm{d}u^2} b(u, X_u) \mathrm{d}u + \frac{t_{k+1} - t}{t_{k+1} - t_k} \int_{t_k}^{t} \mathrm{d}s \int_{s}^{t} \frac{\mathrm{d}^2}{\mathrm{d}u^2} b(u, X_u) \mathrm{d}u.$$

So,

$$\text{(iv)} \leq (t - t_k)(t_{k+1} - t_k) \int_{t_k}^{t_{k+1}} \left\| \frac{\mathrm{d}^2}{\mathrm{d}s^2} b(s, X_s) \right\| \mathrm{d}s.$$

The proof is finished by directly combining the four parts. □

By Lemma E.2, the bias error

$$A \lesssim L(t - t_k)\varepsilon_{1,k}(X_{t_k}) + L(t - t_k)\varepsilon_{1,k+1}(X_{t_{k+1}})$$
$$+ L(t - t_k) \int_{t_k}^{t_{k+1}} \left\| \frac{\mathrm{d}}{\mathrm{d}s} b(s, X_s) \right\| \mathrm{d}s$$
$$+ (t - t_k)Lh_k \int_{t_k}^{t_{k+1}} \left\| \frac{\mathrm{d}^2}{\mathrm{d}s^2} b(s, X_s) \right\| \mathrm{d}s.$$

For the estimation error $B$, clearly,

$$\|B\| \leq \frac{t_{k+1} - t}{t_{k+1} - t_k}\varepsilon_{1,k}(X_{t_k}) + \frac{t - t_k}{t_{k+1} - t_k}\varepsilon_{1,k+1}(X_{t_{k+1}}).$$

For the discretization error $C$, according to the proof of Lemma E.2, we obtain that

$$\|C\| \lesssim (t_{k+1} - t_k) \int_{t_k}^{t_{k+1}} \left\| \frac{\mathrm{d}^2}{\mathrm{d}s^2} b(s, X_s) \right\| \mathrm{d}s.$$

Combine the above three terms, and apply Lemmas F.13 and F.15. Then, we get:

$$\mathbb{E}_{X_t \sim \rho(t)} \left[ \|\tilde{b}(t, X_t) - b(t, X_t)\| \cdot \|\nabla \ln \rho(t, X_t)\| \right]$$
$$\lesssim \mathbb{E}_{X_t \sim \rho(t)} \left[ \|\tilde{b}(t, X_t)\|^{6/5} \right]^{5/6} \cdot \mathbb{E}_{X_t \sim \rho(t)} \left[ \|s(t, X_t)\|^6 \right]^{1/6}$$
$$\lesssim \gamma(t)^{-1} d^{1/2} \left[ \mathbb{E} \left[ \|\varepsilon_{1,k}(X_{t_k})\|^2 \right]^{1/2} + \mathbb{E} \left[ \|\varepsilon_{1,k}(X_{t_{k+1}})\|^2 \right]^{1/2} \right]$$
$$+ \gamma(t)^{-1} d^{1/2} Lh_k \left[ (t - t_k)^{1/5} \int_{t_k}^t \left\| \frac{\mathrm{d}}{\mathrm{d}s} b(s, X_s) \right\|^{6/5} \mathrm{d}s \right]^{5/6}$$
$$+ \gamma(t)^{-1} d^{1/2} h_k \left[ (t - t_k)^{1/5} \int_{t_k}^t \left\| \frac{\mathrm{d}^2}{\mathrm{d}s^2} b(s, X_s) \right\|^{6/5} \mathrm{d}s \right]^{5/6}$$
$$\lesssim \gamma(t)^{-1} d^{1/2} \left[ \mathbb{E} \left[ \varepsilon_{1,k}(X_{t_k})^2 \right]^{1/2} + \mathbb{E} \left[ \varepsilon_{1,k}(X_{t_{k+1}})^2 \right]^{1/2} \right]$$
$$+ \bar{\gamma}_k^{-3} d^{3/2} h_k^2 \mathbb{E} \left[ \|x_0 - x_1\|^6 \right]^{1/2} + d^3 \bar{\gamma}_k^{-6}$$
$$+ L\bar{\gamma}_k^{-2} d\mathbb{E} \left[ \|x_0 - x_1\|^6 \right]^{1/3} + Ld^2 \bar{\gamma}_k^{-4}.$$

Integrate and write $M = \max\{\mathbb{E} \left[ \|x_0 - x_1\|^6 \right]^{1/3}, L, d\}$, then

$$\int_{t_0}^{t_N} \mathbb{E}_{X_t \sim \rho(t)} \left[ \|\tilde{b}(t, X_t) - b(t, X_t)\| \cdot \|\nabla \ln \rho(t, X_t)\| \right] \mathrm{d}t$$
$$\lesssim d^{1/2} \varepsilon_{\mathrm{drift}} S(\gamma, t_0, t_N)^{1/2} + \sum_{k=0}^{N-1} h_k^3 \left[ \bar{\gamma}_k^{-6} d^3 + M^3 \bar{\gamma}_k^{-4} \right].$$

### E.3 CONTROLLING THE DIVERGENCE ERROR

Use the notation $\varepsilon_{2,k}(x) = \|\nabla \hat{b}(t_k, x) - \nabla b(t_k, x)\|_F$. First,

$$\nabla \cdot \tilde{b}(t, X_t) - \nabla \cdot b(t, X_t) = \mathrm{tr} \left[ \frac{t_{k+1} - t}{t_{k+1} - t_k} \nabla \hat{b}(t_k, z) \nabla G_{t_k \to t}(z)^{-1} \right.$$
$$+ \frac{t - t_k}{t_{k+1} - t_k} \nabla \hat{b}(t_{k+1}, F_{t_k \to t_{k+1}}(z)) \nabla F_{t_k \to t_{k+1}}(z) \nabla G_{t_k \to t}(z)^{-1}$$
$$\left. - \nabla b(t, X_t) \nabla G_{t_k \to t}(z) \nabla G_{t_k \to t}(z)^{-1} \right].$$

Then, by Lemma E.1,

$$
\begin{aligned}
\left| \nabla \cdot \tilde{b}(t, X_t) - \nabla \cdot b(t, X_t) \right| &\lesssim \left| \mathrm{tr}\left[ \frac{t_{k+1} - t}{t_{k+1} - t_k} \nabla \hat{b}(t_k, z) \right. \right. \\
&\qquad + \frac{t - t_k}{t_{k+1} - t_k} \nabla \hat{b}(t_{k+1}, F_{t_k \to t_{k+1}}(z)) \nabla F_{t_k \to t_{k+1}}(z) \\
&\qquad \left. \left. - \nabla b(t, X_t) \nabla G_{t_k \to t}(z) \right] \right| \\
&= \left| \mathrm{tr}\left[ D_{\mathrm{bias}} + D_{\mathrm{est}} + D_{\mathrm{dis}} \right] \right|,
\end{aligned}
$$

where the terms are defined as:

$$
\begin{aligned}
D_{\mathrm{bias}} &= \frac{t_{k+1} - t}{t_{k+1} - t_k} \left[ \nabla \hat{b}(t_k, z) - \nabla \hat{b}(t_k, X_{t_k}) \right] \\
&\quad + \frac{t - t_k}{t_{k+1} - t_k} \left[ \nabla \hat{b}(t_{k+1}, F_{t_k \to t_{k+1}}(z)) \nabla F_{t_k \to t_{k+1}}(z) - \nabla \hat{b}(t_{k+1}, X_{t_{k+1}}) \nabla G_{t_k \to t_{k+1}}(X_{t_k}) \right] \\
&\quad - \left[ \nabla b(t, X_t) \nabla G_{t_k \to t}(z) - \nabla b(t, X_t) \nabla G_{t_k \to t}(X_{t_k}) \right], \\
D_{\mathrm{est}} &= \frac{t_{k+1} - t}{t_{k+1} - t_k} \left[ \nabla \hat{b}(t_k, X_{t_k}) - \nabla b(t_k, X_{t_k}) \right] \\
&\quad + \frac{t - t_k}{t_{k+1} - t_k} \left[ \nabla \hat{b}(t_{k+1}, X_{t_{k+1}}) \nabla G_{t_k \to t_{k+1}}(X_{t_k}) - \nabla b(t_{k+1}, X_{t_{k+1}}) \nabla G_{t_k \to t_{k+1}}(X_{t_k}) \right], \\
D_{\mathrm{dis}} &= \frac{t_{k+1} - t}{t_{k+1} - t_k} \nabla b(t_k, X_{t_k}) \nabla G_{t_k \to t_k}(X_{t_k}) + \frac{t - t_k}{t_{k+1} - t_k} \nabla b(t_{k+1}, X_{t_{k+1}}) \nabla G_{t_k \to t_{k+1}}(X_{t_k}) \\
&\quad - \nabla b(t, X_t) \nabla G_{t_k \to t}(X_{t_k}).
\end{aligned}
$$

**Bias Error**   For $D_{\mathrm{bias}}$, with Assumption 4.4,

$$
\begin{aligned}
|\mathrm{tr}[D_{\mathrm{bias}}]| &\leq \frac{t_{k+1} - t}{t_{k+1} - t_k} L^{3/2} \|z - X_{t_k}\| \\
&\quad + \frac{t - t_k}{t_{k+1} - t_k} \left\| \nabla \hat{b}(t_{k+1}, F_{t_k \to t_{k+1}}(z)) - \nabla \hat{b}(t_{k+1}, X_{t_{k+1}}) \right\|_F \cdot \| \nabla F_{t_k \to t_{k+1}}(z) \|_F \\
&\quad + \frac{t - t_k}{t_{k+1} - t_k} \| \nabla \hat{b}(t_{k+1}, X_{t_{k+1}}) \|_F \cdot \left\| \nabla F_{t_k \to t_{k+1}}(z) - \nabla G_{t_k \to t_{k+1}}(X_{t_k}) \right\|_F \\
&\quad + \| \nabla b(t, X_t) \|_F \cdot \| \nabla G_{t_k \to t}(X_{t_k}) - \nabla G_{t_k \to t}(z) \|_F \\
&\overset{(a)}{\lesssim} \frac{t_{k+1} - t}{t_{k+1} - t_k} L^{3/2} \|z - X_{t_k}\| + \frac{t - t_k}{t_{k+1} - t_k} L^{3/2} \left\| F_{t_k \to t_{k+1}}(z) - X_{t_{k+1}} \right\| \\
&\quad + \frac{t - t_k}{t_{k+1} - t_k} L \left\| h_k \nabla \hat{b}(t_k, z) - \frac{1}{2} h_k \nabla \hat{b}(t_k, X_{t_k}) \right. \\
&\quad \qquad \left. - \frac{1}{2} h_k \nabla \hat{b}(t_{k+1}, F_{t_k \to t_{k+1}}(X_{t_k})) \nabla F_{t_k \to t_{k+1}}(X_{t_k}) \right\|_F \\
&\quad + \| \nabla b(t, X_t) \|_F \cdot L^{3/2} h_k \| X_{t_k} - z \|
\end{aligned}
$$

$$
\overset{(b)}{\lesssim} \|z - X_{t_k}\| L^{3/2} \left(1 + h_k \|\nabla b(t, X_t)\|_F\right) + L^{3/2} \|F_{t_k \to t_{k+1}}(X_{t_k}) - X_{t_{k+1}}\|
$$
$$
+ L h_k \|\nabla F_{t_k \to t_{k+1}}(X_{t_k}) - I_d\| \cdot \|\nabla \hat{b}(t_{k+1}, X_{t_{k+1}})\|_F
$$
$$
+ L h_k \left\|\nabla \hat{b}(t_k, X_{t_k}) - \nabla \hat{b}(t_{k+1}, X_{t_{k+1}})\right\|_F
$$
$$
\overset{(c)}{\lesssim} L^{3/2}(1 + h_k \|\nabla b(t, X_t)\|_F) \left[ h_k \varepsilon_{1,k}(X_{t_k}) + h_k \varepsilon_{1,k+1}(X_{t_{k+1}}) \right.
$$
$$
+ h_k^2 L \int_{t_k}^{t_{k+1}} \left\|\frac{\mathrm{d}}{\mathrm{d}s} b(s, X_s)\right\| \mathrm{d}s + h_k^2 \int_{t_k}^{t_{k+1}} \left\|\frac{\mathrm{d}^2}{\mathrm{d}s^2} b(s, X_s)\right\| \mathrm{d}s \right]
$$
$$
+ L^{3/2} h_k \int_{t_k}^{t_{k+1}} \left\|\frac{\mathrm{d}}{\mathrm{d}s} b(s, X_s)\right\| \mathrm{d}s
$$
$$
+ L h_k \left(\varepsilon_{2,k}(X_{t_k}) + \varepsilon_{2,k+1}(X_{t_{k+1}})\right)
$$
$$
+ L h_k \int_{t_k}^{t_{k+1}} \left\|\frac{\mathrm{d}}{\mathrm{d}s} \left(\nabla b(s, X_s)\right)\right\|_F \mathrm{d}s.
$$

The inequality (a) applies Assumption 4.4; the inequality (b) rearranges the terms and applies triangle inequalities; the inequality (c) expands the differences into the form of integrals by Lemma D.3, E.2, and we control the difference $\left\|\nabla \hat{b}(t_k, X_{t_k}) - \nabla \hat{b}(t_{k+1}, X_{t_{k+1}})\right\|_F$ by controlling $\left\|\nabla b(t_k, X_{t_k}) - \nabla b(t_{k+1}, X_{t_{k+1}})\right\|_F$.

Notably, in the first term, when $d h_k \bar{\gamma}_k^{-2} \lesssim 1$ and $h_k d^{1/2} \bar{\gamma}_k^{-1} \mathbb{E}\left[\|x_0 - x_1\|^p\right]^{1/p} \lesssim 1$ ($p > 1$ is a small constant), we have $h_k \mathbb{E}\|\nabla b(t, X_t)\|^p]^{1/p} \lesssim 1$, so based on the assumptions on $h_k$, Lemmas F.13, F.15, F.14 and Hölder's inequality,

$$
\mathbb{E}\left[|\mathrm{tr}[D_{\mathrm{bias}}]|\right] \lesssim L^{3/2} h_k^2 \left(\mathbb{E}\left[\|x_0 - x_1\|^6\right]^{1/3} \bar{\gamma}_k^{-1} d^{1/2} + \bar{\gamma}^{-3} d^2\right)
$$
$$
+ L^{3/2} h_k^3 \left[\mathbb{E}\left[\|x_0 - x_1\|^6\right]^{1/2} d \bar{\gamma}_k^{-2} + d^{5/2} \bar{\gamma}_k^{-5}\right]
$$
$$
+ L h_k^2 \left[\mathbb{E}\left[\|x_0 - x_1\|^6\right]^{1/3} d \bar{\gamma}_k^{-2} + d^2 \bar{\gamma}_k^{-4}\right]
$$
$$
+ L^{3/2} h_k \left(\mathbb{E}[\varepsilon_{1,k}(X_{t_k})] + \mathbb{E}[\varepsilon_{1,k+1}(X_{t_{k+1}})]\right)
$$
$$
+ L h_k \left(\mathbb{E}[\varepsilon_{2,k}(X_{t_k})] + \mathbb{E}[\varepsilon_{2,k+1}(X_{t_{k+1}})]\right).
$$

**Estimation Error** For $D_{\mathrm{est}}$,

$$
|\mathrm{tr}[D_{\mathrm{estimate}}]| \leq \frac{t_{k+1} - t}{t_{k+1} - t_k} \varepsilon_{2,k}(X_{t_k}) + \frac{t - t_k}{t_{k+1} - t_k} \varepsilon_{2,k+1}(X_{t_{k+1}}) \cdot \|\nabla G_{t_k \to t_{k+1}}(X_{t_k})\|
$$
$$
\lesssim \varepsilon_{2,k}(X_{t_k}) + \varepsilon_{2,k+1}(X_{t_{k+1}}).
$$

**Discretization Error** For $D_{\mathrm{dis}}$, note that $\|\nabla G_{t_k \to t}(X_{t_k}) - I_d\|$ is of order $O(h)$, we cannot use this approximation because we are expecting the error of order $O(h^2)$. Thus we use a different approach to deal with this term. By taking the second-order derivative,

$$
|\mathrm{tr}[D_{\mathrm{dis}}]| \leq (t_{k+1} - t_k) \int_{t_k}^{t_{k+1}} \left|\frac{\mathrm{d}^2}{\mathrm{d}s^2} \mathrm{tr}\left[\nabla b(s, X_s) \nabla G_{t_k \to s}(X_{t_k})\right]\right| \mathrm{d}s.
$$

We now explicitly write the derivatives of $\nabla G_{t_k \to s}$ below for further discussion:

$$
\nabla G_{t_k \to t}(x) = I_d + \left[(t - t_k) - \frac{(t - t_k)^2}{2h_k}\right] \nabla \hat{b}(t_k, x)
$$
$$
+ \frac{(t - t_k)^2}{2h_k} \nabla \hat{b}(t_{k+1}, F_{t_k \to t_{k+1}}(x)) \cdot (I_d + h_k \nabla \hat{b}(t_k, x)),
$$

$$\frac{\mathrm{d}}{\mathrm{d}t}\nabla G_{t_k \to t}(x) = \frac{t_{k+1} - t}{h_k}\nabla \hat{b}(t_k, x)$$

$$+ \frac{t - t_k}{h_k}\nabla \hat{b}(t_{k+1}, F_{t_k \to t_{k+1}}(x)) \cdot (I_d + h_k \nabla \hat{b}(t_k, x)),$$

$$\frac{\mathrm{d}^2}{\mathrm{d}t^2}\nabla G_{t_k \to t}(x) = \frac{1}{h_k}\left[\nabla \hat{b}(t_{k+1}, F_{t_k \to t_{k+1}}(x)) \cdot (I_d + h_k \nabla \hat{b}(t_k, x)) - \nabla \hat{b}(t_k, x)\right].$$

So, by our assumption that $h_k L \lesssim 1$,

$$\|\nabla G_{t_k \to t}(x)\|_F \lesssim 1, \qquad \left\|\frac{\mathrm{d}}{\mathrm{d}t}\nabla G_{t_k \to t}(x)\right\|_F \lesssim L,$$

$$\left\|\frac{\mathrm{d}^2}{\mathrm{d}t^2}\nabla G_{t_k \to t}(X_{t_k})\right\|_F \lesssim L\left(\varepsilon_{2,k}(X_{t_k}) + \|\nabla b(t_k, X_{t_k})\|_F\right)$$

$$+ \frac{L^{3/2}}{h_k}\|F_{t_k \to t_{k+1}}(X_{t_k}) - X_{t_{k+1}}\|_F$$

$$+ \frac{1}{h_k}\int_{t_k}^{t_{k+1}}\left\|\frac{\mathrm{d}}{\mathrm{d}s}\nabla b(s, X_s)\right\|_F \mathrm{d}s$$

$$+ \frac{1}{h_k}\varepsilon_{2,k}(X_{t_k}) + \frac{1}{h_k}\varepsilon_{2,k+1}(X_{t_{k+1}}),$$

The constants omitted by the notation "$\lesssim$" above is uniform for all $x$ (or $X_{t_k}$) and $t$. So,

$$\mathbb{E}\left[\mathrm{tr}[D_{\mathrm{dis}}]\right] \lesssim h_k \int_{t_k}^{t_{k+1}}\left\{\mathbb{E}\left[\left\|\frac{\mathrm{d}^2}{\mathrm{d}s^2}\nabla b(s, X_s)\right\|_F\right]\right.$$

$$+ L\mathbb{E}\left[\left\|\frac{\mathrm{d}}{\mathrm{d}s}\nabla b(s, X_s)\right\|_F\right]$$

$$+ \mathbb{E}\left[\|\nabla b(s, X_s)\|_F \cdot \left\|\frac{\mathrm{d}^2}{\mathrm{d}s^2}\nabla G_{t_k \to s}(X_{t_k})\right\|_F\right]\right\}\mathrm{d}s$$

$$\lesssim h_k\left(\mathbb{E}\left[\varepsilon_{2,k}(X_{t_k})^2\right]^{1/2} + \mathbb{E}\left[\varepsilon_{2,k+1}(X_{t_{k+1}})^2\right]^{1/2} + L^{3/2}h_k\mathbb{E}[\varepsilon_{1,k}(X_{t_k})^2]^{1/2}\right)$$

$$\cdot \left(d\gamma(t)^{-2} + d^{1/2}\gamma(t)^{-1}\mathbb{E}\left[\|x_0 - x_1\|^2\right]^{1/2}\right)$$

$$+ h_k^2\left[\mathbb{E}\left[\|x_0 - x_1\|^6\right]^{1/2}d^{3/2}\bar{\gamma}_k^{-3} + d^3\bar{\gamma}_k^{-6}\right.$$

$$+ (L + d\bar{\gamma}_k^{-2})\left(\mathbb{E}\left[\|x_0 - x_1\|^6\right]^{1/3}d\bar{\gamma}_k^{-2} + \bar{\gamma}_k^{-4}d^2\right)\right]$$

Therefore, by adding $D_{\mathrm{bias}}$, $D_{\mathrm{est}}$ and $D_{\mathrm{dis}}$ together and integrating with respect to $t$, we can obtain that

$$\int_{t_0}^{t_N}\mathbb{E}_{X_t \sim \rho(t)}\left[|\nabla \cdot \tilde{b}(t, X_t) - \nabla \cdot b(t, X_t)|\right]\mathrm{d}t$$

$$\lesssim \sum_{k=0}^{N-1}h_k\left(\mathbb{E}\left[\varepsilon_{2,k}(X_{t_k})^2\right]^{1/2} + \mathbb{E}\left[\varepsilon_{2,k}(X_{t_{k+1}})^2\right]^{1/2}\right)$$

$$+ \sum_{k=0}^{N-1}h_k L^{1/2}\left(\mathbb{E}\left[\varepsilon_{1,k}(X_{t_k})^2\right]^{1/2} + \mathbb{E}\left[\varepsilon_{1,k}(X_{t_{k+1}})\right]\right)$$

$$+ \sum_{k=0}^{N-1}h_k^3\left(d^3\bar{\gamma}^{-6} + M^3\bar{\gamma}^{-4}\right),$$

where we denote $M = \max\left\{\mathbb{E}\left[\|x_0 - x_1\|^6\right]^{1/3}, L, d\right\}$.

Combining the above discussions, we can get the bounds given in Theorem 5.4.

## F  TECHNICAL LEMMAS

In this section, we use the notation $x_t$ to denote the stochastic process

$$x_t = I(t, x_0, x_1) + \gamma(t)z$$

defined in the stochastic interpolant, and use the notation $X_t$ to denote the process that solves the equation

$$\frac{\mathrm{d}}{\mathrm{d}t}X_t = b(t, X_t)$$

and satisfies $X_t \sim \rho(t)$.

Recall that in stochastic interpolants, we have defined $b(t, x) = \mathbb{E}[\partial_t I(t, x_0, x_1) + \dot{\gamma}z | x_t = x]$ and $s(t, x) = \nabla \log \rho(t, x) = -\gamma^{-1}(t)\mathbb{E}[z | x_t = x]$. For convenience, we define $v(t, x) = \mathbb{E}[\partial_t I(t, x_0, x_1) | x_t = x]$.

### F.1  THE CONCRETE FORM OF DERIVATIVES OF $s(t, x)$ AND $v(t, x)$

Before introducing the lemmas, we first define the notation $f_t = -\frac{\|x - I(t, x_0, x_1)\|^2}{2\gamma(t)^2}$, so that the conditional density $p(x_t | x_0, x_1) \propto \exp(f_t)$. The first lemma shows that how the jacobian matrices of $v$ and $s$ can be expressed.

**Lemma F.1.**

$$\nabla_x v(t, x) = \mathrm{Cov}(\partial_t I, \nabla_x f_t | x_t = x),$$
$$\nabla_x s(t, x) = \mathrm{Cov}(\nabla_x f_t, \nabla_x f_t | x_t = x) - \gamma^{-2} I_d.$$

*Proof.*

$$\nabla_x v(t, x) = \nabla_x \frac{\int \exp(f_t)\partial_t I \mathrm{d}\nu}{\int \exp(f_t)\mathrm{d}\nu}.$$

The conditions for Lebesgue dominated convergence theorem can be easily checked, so the order of differential and integral can be alternated. Then

$$\nabla_x v(t, x) = \frac{\int \exp(f_t)(\nabla_x \partial_t I)\mathrm{d}\nu}{\int \exp(f_t)\mathrm{d}\nu} + \frac{\int \exp(f_t)\partial_t I \otimes \nabla_x f_t \mathrm{d}\nu}{\int \exp(f_t)\mathrm{d}\nu}$$
$$- \frac{\int \exp(f_t)\partial_t I \mathrm{d}\nu \otimes \int \exp(f_t)\nabla_x f_t \mathrm{d}\nu}{\left[\int \exp(f_t)\mathrm{d}\nu\right]^2}$$
$$= \mathrm{Cov}(\partial_t I, \nabla_x f_t | x_t = x).$$

Here the notation $\otimes$ represents the tensor product. Similarly,

$$\nabla_x s(t, x) = \frac{\int \exp(f_t)(\nabla_x^2 f_t)\mathrm{d}\nu}{\int \exp(f_t)\mathrm{d}\nu} + \frac{\int \exp(f_t)\nabla_x f_t \otimes \nabla_x f_t \mathrm{d}\nu}{\int \exp(f_t)\mathrm{d}\nu}$$
$$- \frac{\int \exp(f_t)\nabla_x f_t \mathrm{d}\nu \otimes \int \exp(f_t)\nabla_x f_t \mathrm{d}\nu}{\left[\int \exp(f_t)\mathrm{d}\nu\right]^2}$$
$$= \mathrm{Cov}(\nabla_x f_t, \nabla_x f_t | x_t = x) + \nabla_x^2 \left(-\frac{\|x - I\|^2}{2\gamma^2}\right),$$
$$= \mathrm{Cov}(\nabla_x f_t, \nabla_x f_t | x_t = x) - \gamma^{-2} I_d.$$

$\square$

In the above calculations, the key is to change the order of taking differential and integral, and rearrange the final formula into the form of conditional expectations. The higher order derivatives can be obtained similarly, but since the calculation is too long, we will give the results without showing the detailed proof. Lemmas F.2 and F.3 are similar as Lemma F.1, while we consider the second-order and third-order derivatives.

**Lemma F.2.**
$$\nabla_x^2 v(t,x) = \mathbb{E}[(\partial_t I - v(t,x)) \otimes (\nabla_x f_t - s(t,x)) \otimes (\nabla_x f_t - s(t,x))|x_t = x],$$
$$\nabla_x^2 s(t,x) = \mathbb{E}[(\nabla_x f_t - s(t,x)) \otimes (\nabla_x f_t - s(t,x)) \otimes (\nabla_x f_t - s(t,x))|x_t = x],$$

*Note that $v(t,x)$ and $s(t,x)$ are the conditional expectations of $\partial_t I$ and $\nabla_x f_t$, respectively.*

**Lemma F.3.**
$$\nabla_x^3 v(t,x) = \mathbb{E}[(\partial_t I - v(t,x)) \otimes (\nabla_x f_t - s(t,x))^{\otimes 3}|x_t = x]$$
$$- \mathcal{T}(\mathrm{Cov}(\partial_t I, \nabla_x f_t|x_t = x) \otimes \mathrm{Cov}(\nabla_x f_t, \nabla_x f_t|x_t = x)),$$
$$\nabla_x^3 s(t,x) = \mathbb{E}[(\nabla_x f_t - s(t,x))^{\otimes 4}|x_t = x]$$
$$- \mathcal{T}(\mathrm{Cov}(\nabla_x f_t, \nabla_x f_t|x_t = x)^{\otimes 2}).$$

*Here, the notation $A^{\otimes n}$ represents the tensor product of $n$ tensors $A$. The operator $\mathcal{T}$ is defined as*
$$\mathcal{T} : \mathbb{R}^{d \times d \times d \times d} \to \mathbb{R}^{d \times d \times d \times d}, \quad \mathcal{T}(X) = \mathcal{T}_{1234}(X) + \mathcal{T}_{1324}(X) + \mathcal{T}_{1423}(X),$$

*where*
$$[\mathcal{T}_{p_1 p_2 p_3 p_4}(X)]_{i_1 i_2 i_3 i_4} = X_{i_{p_1} i_{p_2} i_{p_3} i_{p_4}}.$$

Similarly to the derivatives with respect to $x$, the following lemmas (Lemmas F.4, F.5 and F.6) calculates the derivatives of $v$, $s$ and their Jacobian matrices with respect to the time $t$.

**Lemma F.4.**
$$\partial_t v(t,x) = \mathrm{Cov}[\partial_t I, \partial_t f_t|x_t = x] + \mathbb{E}[\partial_t^2 I|x_t = x],$$
$$\partial_t s(t,x) = \mathrm{Cov}[\nabla_x f_t, \partial_t f_t|x_t = x] + \mathbb{E}[\partial_t \nabla_x f_t|x_t = x],$$
$$\partial_t \nabla_x v(t,x) = \mathrm{Cov}(\partial_t^2 I, \nabla_x f_t|x_t = x) + \mathrm{Cov}(\partial_t I, \partial_t \nabla_x f_t|x_t = x)$$
$$+ \mathbb{E}[(\partial_t I - v(t,x)) \otimes (\nabla_x f_t - s(t,x)) \otimes (\partial_t f_t - m_{\partial_t f_t})|x_t = x],$$
$$\partial_t \nabla_x s(t,x) = \mathrm{Cov}(\partial_t \nabla_x f_t, \nabla_x f_t|x_t = x) + \mathrm{Cov}(\nabla_x f_t, \partial_t \nabla_x f_t|x_t = x)$$
$$+ \mathbb{E}[(\nabla_x f_t - s(t,x)) \otimes (\nabla_x f_t - s(t,x)) \otimes (\partial_t f_t - m_{\partial_t f_t})|x_t = x].$$

*In the above formula, we use the notation $m_V = \mathbb{E}[V|x_t = x]$ for simplicity.*

**Lemma F.5.**
$$\partial_t^2 v(t,x) = \mathbb{E}[\partial_t^3 I|x_t = x] + 2\mathrm{Cov}[\partial_t^2 I, \partial_t f_t|x_t = x] + \mathrm{Cov}[\partial_t I, \partial_t^2 f_t|x_t = x]$$
$$+ \mathbb{E}[(\partial_t I - v(t,x)) \otimes (\partial_t f_t - m_{\partial_t f_t}) \otimes (\partial_t f_t - m_{\partial_t f_t})|x_t = x],$$
$$\partial_t^2 s(t,x) = \mathbb{E}[\partial_t^2 \nabla_x f_t|x_t = x] + 2\mathrm{Cov}[\partial_t \nabla_x f_t, \partial_t f_t|x_t = x] + \mathrm{Cov}[\nabla_x f_t, \partial_t^2 f_t|x_t = x]$$
$$+ \mathbb{E}[(\nabla_x f_t - s(t,x)) \otimes (\partial_t f_t - m_{\partial_t f_t}) \otimes (\partial_t f_t - m_{\partial_t f_t})|x_t = x],$$

**Lemma F.6.**
$$\partial_t^2 \nabla_x v(t,x) = \mathrm{Cov}(\partial_t^3 I, \nabla_x f_t|x_t = x) + 2\mathrm{Cov}(\partial_t^2 I, \partial_t \nabla_x f_t|x_t = x)$$
$$+ \mathrm{Cov}(\partial_t I, \partial_t^2 \nabla_x f_t|x_t = x)$$
$$+ 2\mathbb{E}[(\partial_t^2 I - m_{\partial_t^2 I}) \otimes (\nabla_x f_t - s(t,x)) \otimes (\partial_t f_t - m_{\partial_t f_t})|x_t = x]$$
$$+ 2\mathbb{E}[(\partial_t I - v(t,x)) \otimes (\partial_t \nabla_x f_t - m_{\partial_t \nabla_x f_t}) \otimes (\partial_t f_t - m_{\partial_t f_t})|x_t = x]$$
$$+ \mathbb{E}[(\partial_t I - v(t,x)) \otimes (\nabla_x f_t - m_{\nabla_x f_t}) \otimes (\partial_t^2 f_t - m_{\partial_t^2 f_t})|x_t = x]$$
$$+ \mathbb{E}[(\partial_t I - v(t,x)) \otimes (\partial_t f_t - m_{\partial_t f_t})^{\otimes 3}|x_t = x]$$
$$- \mathcal{T}(\mathrm{Cov}(\partial_t I, \partial_t f_t|x_t = x) \otimes \mathrm{Cov}(\partial_t f_t, \partial_t f_t|x_t = x))$$
$$\partial_t^2 \nabla_x s(t,x) = \mathrm{Cov}(\partial_t^2 \nabla_x f_t, \nabla_x f_t|x_t = x) + 2\mathrm{Cov}(\partial_t \nabla_x f_t, \partial_t \nabla_x f_t|x_t = x)$$
$$+ \mathrm{Cov}(\nabla_x f_t, \partial_t^2 \nabla_x f_t|x_t = x)$$
$$+ 2\mathbb{E}[(\partial_t \nabla_x f_t - m_{\partial_t \nabla_x f_t}) \otimes (\nabla_x f_t - s(t,x)) \otimes (\partial_t f_t - m_{\partial_t f_t})|x_t = x]$$
$$+ 2\mathbb{E}[(\nabla_x f_t - s(t,x)) \otimes (\partial_t \nabla_x f_t - m_{\partial_t \nabla_x f_t}) \otimes (\partial_t f_t - m_{\partial_t f_t})|x_t = x]$$
$$+ \mathbb{E}[(\nabla_x f_t - s(t,x)) \otimes (\nabla_x f_t - m_{\nabla_x f_t}) \otimes (\partial_t^2 f_t - m_{\partial_t^2 f_t})|x_t = x]$$
$$+ \mathbb{E}[(\nabla_x f_t - s(t,x)) \otimes (\partial_t f_t - m_{\partial_t f_t})^{\otimes 3}|x_t = x]$$
$$- \mathcal{T}(\mathrm{Cov}(\nabla_x f_t, \partial_t f_t|x_t = x) \otimes \mathrm{Cov}(\partial_t f_t, \partial_t f_t|x_t = x)).$$

### F.2 Upper Bounds on the Derivatives

We first provide a uniform upper bound to explain why Assumption 4.4 is reasonable.

**Lemma F.7.** *Suppose that for all $t \in [0,1]$, $P(\|I(t, x_0, x_1)\| \le R) = 1$, i.e. the data is bounded. Then under Assumption 1,*

$$\|\nabla b(t, x)\|_F \lesssim \gamma(t)^{-4} R^2,$$
$$\|\nabla^2 b(t, x)\|_F \lesssim \gamma(t)^{-6} R^3,$$
$$\|\nabla(\nabla \cdot b(t, x))\| \lesssim \gamma(t)^{-6} R^3.$$

*Proof.* According to Lemma F.1, since $\gamma\dot\gamma = O(1)$, for any $u \in \mathbb{R}^d$, by noticing that $\|x \otimes y\|_F = \|x\| \cdot \|y\|$,

$$\|\nabla b(t, x)\|_F = \|\nabla v(t, x) + \gamma(t)\dot\gamma(t)\nabla s(t, x)\|$$
$$\lesssim \mathbb{E}[(\|(\partial_t I - v(t, x))\| + \|\nabla_x f_t - s(t, x)\|) \cdot \|\nabla_x f_t - s(t, x)\||x_t = x].$$

Since $\|\partial_t I\| \lesssim R$, $\nabla_x f_t - s(t, x) = -\frac{x-I}{\gamma(t)^2} - \mathbb{E}[-\frac{x-I}{\gamma(t)^2}|x_t = x] = \gamma(t)^{-2}(I - \mathbb{E}[I|x_t = x]) \lesssim \gamma(t)^{-2}R$, we have

$$\|\nabla b(t, x)\| \lesssim \gamma(t)^{-4} R^2.$$

Similarly,

$$\|\nabla^2 b(t, x)\|_F \lesssim (\gamma(t)^{-2}R)^2(1 + \gamma(t)^{-2})R \lesssim \gamma(t)^{-6} R^3.$$
$$\|\nabla(\nabla \cdot b(t, x))\| \lesssim (\gamma(t)^{-2}R)^2(1 + \gamma(t)^{-2})R \lesssim \gamma(t)^{-6} R^3.$$

$\square$

Lemma F.7 justifies the choice of Lipschitz constants in Assumption 4.4, where we set $L$ and $L^{3/2}$ as the Lipschitz constants for $\hat{b}(t, x)$ and its spatial derivative, respectively. When $R = O(\sqrt{d})$, as occurs when data are bounded in each dimension, it follows that $L = O(d)$.

The rest of this section analyzes expectation-based upper bounds, rather than uniform bounds with respect to $x$ and $t$.

**Lemma F.8.** *For a Gaussian random variable $z \sim \mathcal{N}(0, I_d)$, for any constant $p \ge 2$,*

$$\mathbb{E}\left[\|z\|^p\right] \le C(p)d^{p/2},$$

*where the constant $C(p) > 0$ only depends on $p$.*

*Proof.* First, by Jensen's inequality,

$$\|z\|^p = (\|z\|^2)^{p/2} = \left(\sum_{k=1}^d |z_i|^2\right)^{p/2} \le d^{p/2} \sum_{k=1}^d \frac{1}{p}|z_i|^p,$$

so

$$\mathbb{E}\left[\|z\|^p\right] \le d^{p/2}\mathbb{E}|z_1|^p.$$

Here, $C(p) = \mathbb{E}|z_1|^p < \infty$ is a constant that only depends on $p$ since $z_1 \sim N(0, 1)$ is a standard 1-dimensional Gaussian variable. $\square$

The following lemmas (Lemma F.9 to Lemma F.12) provide upper bounds on the moments of the time and spatial derivatives of $v$ and $s$. Notably, for these lemmas, the expectation is taken over $x_t \sim \rho(t)$, where recall that $x_t$ is the stochastic interpolant.

**Lemma F.9.** *Under Assumption 4.1, for $p \ge 2$,*

$$\mathbb{E}\left[\|v(t, x_t)\|^p\right] \lesssim \mathbb{E}\left[\|x_0 - x_1\|^p\right],$$
$$\mathbb{E}\left[\|s(t, x_t)\|^p\right] \lesssim \gamma(t)^{-p} d^{p/2}.$$

*Proof.* By law of total probability and Jensen's inequality,

$$\mathbb{E}\left[\|v(t,x_t)\|^p\right] \leq \mathbb{E}\left[\|\partial_t I\|^p\right] \lesssim \mathbb{E}\left[\|x_0 - x_1\|^p\right],$$
$$\mathbb{E}\left[\|s(t,x_t)\|^p\right] \leq \gamma(t)^{-p}\mathbb{E}\left[\|z\|^p\right] \lesssim \gamma(t)^{-p}d^{p/2}.$$

$\square$

**Lemma F.10.** *Under Assumption 4.1, for any $p \geq 1$, we have*

$$\mathbb{E}\left[\|\nabla_x v(t,x_t)\|_F^p\right] \lesssim \gamma(t)^{-p}d^{p/2}\sqrt{\mathbb{E}\left[\|x_0 - x_1\|^{2p}\right]},$$
$$\mathbb{E}\left[\|\nabla_x s(t,x_t)\|_F^p\right] \lesssim \gamma(t)^{-2p}d^p.$$

*Proof.*

$$\|\mathrm{Cov}(\partial_t I, \nabla_x f_t | x_t = x)\|_F \leq \sqrt{\mathbb{E}\left[\|\partial_t I\|^2 | x_t = x\right]\mathbb{E}\left[\|\nabla_x f_t\|^2 | x_t = x\right]}.$$

Then, by Jensen's inequalty and Cauchy-Schwarz inequality,

$$\mathbb{E}\left[\|\nabla_x v(t,x_t)\|_F^p\right] \lesssim \sqrt{\mathbb{E}\left[\|\partial_t I\|^{2p}\right]}\sqrt{\mathbb{E}\left[\|\nabla_x f_t\|^{2p}\right]}$$
$$\lesssim \gamma(t)^{-p}d^{p/2}\sqrt{\mathbb{E}\left[\|x_0 - x_1\|^{2p}\right]}.$$

Similarly,

$$\mathbb{E}\left[\|\nabla_x s(t,x_t)\|_F^p\right] \lesssim \gamma(t)^{-2p}d^{p/2} + \mathbb{E}\left[\|\nabla_x f_t\|^{2p}\right]$$
$$\lesssim \gamma(t)^{-2p}d^p.$$

$\square$

**Lemma F.11.** *Under Assumption 4.1, for any $p \geq 1$,*

$$\mathbb{E}\left[\|\partial_t v(t,x_t)\|^p\right] \lesssim \mathbb{E}\left[\|x_0 - x_1\|^{2p}\right]\gamma^{-p}d^{p/2} + \mathbb{E}\left[\|x_0 - x_1\|^{2p}\right]^{1/2}\gamma^{-2p}d^p,$$
$$\mathbb{E}\left[\|\partial_t s(t,x_t)\|^p\right] \lesssim \gamma^{-3p}d^{3p/2} + \mathbb{E}\left[\|x_0 - x_1\|^{2p}\right]^{1/2}\gamma^{-2p}d^p.$$

**Lemma F.12.** *Under Assumption 4.1, for any $p \geq 1$,*

$$\mathbb{E}\left[|\partial_t \nabla \cdot v(t,x_t)|^p\right] \lesssim \gamma^{-3p}d^{3p/2}\mathbb{E}\left[\|x_0 - x_1\|^{3p}\right]^{1/3} + \gamma^{-2p}d^p\mathbb{E}\left[\|x_0 - x_1\|^{3p}\right]^{2/3},$$
$$\mathbb{E}\left[|\partial_t \nabla \cdot s(t,x_t)|^p\right] \lesssim \gamma^{-4p}d^{2p} + \gamma^{-3p}d^{3p/2}\mathbb{E}\left[\|x_0 - x_1\|^{3p}\right]^{1/3}.$$

The proofs for the above two lemmas are omitted as they almost repeat the proof of Lemma F.10.

Now, with the previous lemmas, we are ready to provide upper bounds on the time-derivatives of $b(t, X_t)$ (Lemmas F.13 and F.15) and its divergence $\nabla \cdot b(t, X_t)$ (Lemmas F.14 and F.16), where $X_t$ is the true solution of the ODE $\mathrm{d}X_t = b(t, X_t)$ that satisfies $X_t \sim \rho(t)$. These results are later used to control the discretization error of numerical methods.

**Lemma F.13.** *Under Assumption 4.1, for $p \geq 1$,*

$$\mathbb{E}\left[\left\|\frac{\mathrm{d}}{\mathrm{d}t}b(t,X_t)\right\|^p\right] \lesssim \mathbb{E}\left[\|x_0 - x_1\|^{3p}\right]^{2/3}\gamma(t)^{-p}d^{p/2} + \gamma(t)^{-3p}d^{3p/2}.$$

*Proof.*

$$\frac{\mathrm{d}}{\mathrm{d}t}b(t,X_t) = \partial_t b(t,X_t) + \nabla_x b(t,X_t)\cdot b(t,X_t)$$
$$= \partial_t v(t,X_t) + \partial_t\left[\dot{\gamma}\gamma s(t,X_t)\right] + \left[\nabla_x v(t,X_t) + \dot{\gamma}\gamma\nabla_x s(t,X_t)\right]\cdot b(t,X_t).$$

By Jensen's inequality, $(\sum_{k=1}^{n} a_i)^p \leq n^{p-1} \sum_{k=1}^{n} |a_i|^p$, so

$$\mathbb{E}\left[\left\|\frac{\mathrm{d}}{\mathrm{d}t}b(t, X_t)\right\|^p\right] \lesssim \mathbb{E}\left[\|\partial_t v(t, X_t)\|^p\right]$$

$$+ \mathbb{E}\left[\|\partial_t s(t, X_t)\|^p\right] + \mathbb{E}\left[\|s(t, X_t)\|^p\right]$$

$$+ \mathbb{E}\left[\|\nabla_x v(t, X_t) + \nabla_x s(t, X_t)\|^p \cdot \|b(t, X_t)\|^p\right]$$

$$\overset{(a)}{\lesssim} \mathbb{E}\left[\|\partial_t v(t, X_t)\|^p\right]$$

$$+ \mathbb{E}\left[\|\partial_t s(t, X_t)\|^p\right] + \mathbb{E}\left[\|s(t, X_t)\|^p\right]$$

$$+ \mathbb{E}\left[\|\nabla_x v(t, X_t) + \nabla_x s(t, X_t)\|^{3p/2}\right]^{2/3} \cdot \mathbb{E}\left[\|b(t, X_t)\|^{3p}\right]^{1/3}$$

$$\overset{(b)}{\lesssim} \mathbb{E}\left[\|x_0 - x_1\|^{3p}\right]^{2/3} \gamma(t)^{-p} d^{p/2} + \gamma(t)^{-3p} d^{3p/2}.$$

The inquality (a) uses Hölder's inequality, while the inequality (b) uses the results of previous lemmas. □

**Lemma F.14.** *Under Assumption 4.1, for $p \geq 1$,*

$$\mathbb{E}\left[\left|\frac{\mathrm{d}}{\mathrm{d}t}(\nabla \cdot b(t, X_t))\right|^p\right] \lesssim \mathbb{E}\left[\|x_0 - x_1\|^{3p}\right]^{2/3} \gamma^{-2p} d^p + \gamma^{-4p} d^{2p},$$

$$\mathbb{E}\left[\left\|\frac{\mathrm{d}}{\mathrm{d}t}\nabla b(t, X_t)\right\|^p\right] \lesssim \mathbb{E}\left[\|x_0 - x_1\|^{3p}\right]^{2/3} \gamma^{-2p} d^p + \gamma^{-4p} d^{2p}.$$

*Proof.* Similarly to the proof of Lemma F.13,

$$\mathbb{E}\left[\left|\frac{\mathrm{d}}{\mathrm{d}t}(\nabla \cdot b(t, X_t))\right|^p\right] \lesssim \mathbb{E}\left[|\mathrm{tr}\,(\partial_t \nabla_x b(t, x))|^p\right]$$

$$+ \mathbb{E}\left[\left|\mathrm{tr}\,(\nabla^2 b(t, X_t)[b(t, X_t)])\right|^p\right].$$

For the first term on the right hand side, we can use Lemma F.12. For the second term, note that for any four vectors $x, y, z, w$, we have

$$|\mathrm{tr}\,[(x \otimes y \otimes z)[w]]| = |\mathrm{tr}\,[(x \otimes y) \cdot (z \cdot w)]| \leq \|x\| \cdot \|y\| \cdot \|z\| \cdot \|w\|.$$

Hence, by the conditional expectation form in Lemma F.2,

$$\mathbb{E}\left[\left|\mathrm{tr}\,(\nabla^2 b(t, X_t)[b(t, X_t)])\right|^p\right] \lesssim \mathbb{E}\left[\|x_0 - x_1\|^{3p}\right]^{2/3} \gamma^{-2p} d^p + \gamma^{-4p} d^{2p}.$$

So,

$$\mathbb{E}\left[\left|\frac{\mathrm{d}}{\mathrm{d}t}(\nabla \cdot b(t, X_t))\right|^p\right] \lesssim \mathbb{E}\left[\|x_0 - x_1\|^{3p}\right]^{2/3} \gamma^{-2p} d^p + \gamma^{-4p} d^{2p}.$$

The case for the Jacobian matrix is the same except where we use $\|x \otimes y\|_F = \|x\| \cdot \|y\|$ instead of the inequality for the trace. □

**Lemma F.15.** *Under Assumption 5.1, for $p \geq 1$,*

$$\mathbb{E}\left[\left\|\frac{\mathrm{d}^2}{\mathrm{d}t^2}b(t, X_t)\right\|^p\right] \lesssim \mathbb{E}\left[\|x_0 - x_1\|^{5p}\right]^{3/5} d^p \gamma(t)^{-2p} + d^{5p/2} \gamma(t)^{-5p}.$$

*Proof.*

$$\frac{\mathrm{d}^2}{\mathrm{d}t^2}b(t, X_t) = \partial_t^2 b(t, X_t) + 2\partial_t \nabla b(t, X_t) \cdot b(t, X_t) + \nabla b(t, X_t) \cdot \partial_t b(t, X_t)$$

$$+ \nabla^2 b(t, X_t)\left[b(t, X_t)^{\otimes 2}\right] + [\nabla b(t, X_t)]^2 b(t, X_t)$$

Consider Lemma F.1 to F.5 where the conditional expectation forms of the above terms are given, then use Hölder's inequality and Jensen's inequality to obtain the upper bound in the Lemma. □

**Lemma F.16.** *Under Assumption 5.1,*

$$\mathbb{E}\left[\left|\frac{\mathrm{d}^2}{\mathrm{d}t^2}\left(\nabla \cdot b(t, X_t)\right)\right|\right] \lesssim \mathbb{E}\left[\|x_0 - x_1\|^6\right]^{1/2} d^{3/2}\gamma(t)^{-3} + d^3\gamma(t)^{-6},$$

$$\mathbb{E}\left[\left\|\frac{\mathrm{d}^2}{\mathrm{d}t^2}\nabla b(t, X_t)\right\|_F\right] \lesssim \mathbb{E}\left[\|x_0 - x_1\|^6\right]^{1/2} d^{3/2}\gamma(t)^{-3} + d^3\gamma(t)^{-6}.$$

*Proof.*

$$\frac{\mathrm{d}^2}{\mathrm{d}t^2}\nabla b(t, X_t) = \partial_t^2 \nabla b(t, X_t) + 2\partial_t \nabla^2 b(t, X_t)[b(t, X_t)] + \nabla^2 b(t, X_t)[\partial_t b(t, X_t)]$$
$$+ \nabla^3 b(t, X_t)\left[b(t, X_t)^{\otimes 2}\right] + \nabla^2 b(t, X_t) \cdot \nabla b(t, X_t) \cdot b(t, X_t).$$

The rest is similar to the previous lemma. □

