# OpenReview forum: "Finite-Time Convergence Analysis of ODE-based Generative Models for Stochastic Interpolants"
_ICLR.cc/2026/Conference — ICLR 2026 Poster_

### Official Review · Reviewer_9Pnh · 2025-10-30

**Soundness:** 3
**Presentation:** 3
**Contribution:** 2
**Rating:** 6
**Confidence:** 4

**Summary:**

This work provides the first finite-time convergence analysis for ODE-based stochastic interpolants, establishing total variation error bounds for the Euler and Heun methods. The results yield optimized iteration complexity and step-size schedules, offering improved smoothness and dimensional dependence compared to prior diffusion model analyses.

**Strengths:**

(1) The paper is the first to analyze the convergence of ODE-based stochastic interpolants. The analysis is rigorous, and the presentation is very clear. The convergence results show improvement compared with prior works.

(2) It conducts synthetic experiments to exhibit the TV distance.

**Weaknesses:**

(1) There are some typos and ambiguities in the proof.
Line 730: In the triangle inequality, the last term (term C) seems to be $b(t,X_t)$, instead of $b(t,X_{t_k})$. The remaining proof should be correct.

Line 792: It takes me some time to verify the correctness of these inequalities. However, the reasoning skips too much explanation of why they work. I suggest adding a detailed description of why we need these inequalities and how they are proved, including the calculation of Holder coefficients.

Line 927: The $G_{t_k \rightarrow t_{k+1}}$ should be $F$. Moreover, the definition of $F$ should be restated to make it clearer.

(2) At first glance, I feel that the proof technique is very similar to that in Li et al.2025(b) with the application of Lemma C.2. After a careful examination, I find that it contains some novel techniques that should be highlighted. Could you add an additional section (Maybe in the appendix) to discuss the novel techniques (not just bounds) different from those used in Li et al.2025(b), and why these calculations can lead to a better result? In my opinion, this might be very helpful to readers. Same as the analysis of Heun's method, for example, Huang et al. 2025.

**Questions:**

In my understanding, many techniques are closely related and can be applied to previous analysis of diffusion ODEs. Do you think there are any properties specific to the stochastic interpolant problem (I.e., starting from another distribution instead of Gaussian noise)? For instance, can you provide some concrete examples to show the difference between the stochastic interpolant and the original diffusion process?

---

> ### Author Response · Authors · 2025-11-20
>
> Thank you for your time and effort in reviewing our paper! We are grateful for your constructive suggestions, which have significantly guided our improvements. Please find our responses to your comments below.
>
>
> ### Weaknesses
>
> > There are some typos and ambiguities in the proof. Line 730: In the triangle inequality, the last term (term C) seems to be $b(t,X_t)$, instead of $b(t,X_{t_k})$. The remaining proof should be correct.
>
> > Line 792: It takes me some time to verify the correctness of these inequalities. However, the reasoning skips too much explanation of why they work. I suggest adding a detailed description of why we need these inequalities and how they are proved, including the calculation of Holder coefficients.
>
> > Line 927: The $G_{t_k\to t_{k+1}}$ should be $F$. Moreover, the definition of $F$ should be restated to make it clearer.
>
> A: Thank you for your suggestion! We have corrected the typos and improved the clarity of the relevant parts of the proof. Please see Appendix C and D in our revision.
>
> > At first glance, I feel that the proof technique is very similar to that in Li et al.2025(b) with the application of Lemma C.2. After a careful examination, I find that it contains some novel techniques that should be highlighted. Could you add an additional section (Maybe in the appendix) to discuss the novel techniques (not just bounds) different from those used in Li et al.2025(b), and why these calculations can lead to a better result? In my opinion, this might be very helpful to readers. Same as the analysis of Heun's method, for example, Huang et al. 2025.
>
> A: We appreciate your indepth observation regarding the similarity and difference to the techniques used in Li et al. (2025b) and Huang et al. (2025). While our proofs share certain high-level ideas with these works, our analysis also incorporates several technical innovations that lead to sharper results.
>
> First, compared to the method by Li et al. (2025b), we apply a different partition of the drift error between the interpolated process and the true process, where both the estimation error and discretization error appears in the formula. Then, we investigate the smoothness properties of $b(t,x)$ to further control the discretization error. In addition, compared to Huang et al. (2025), by applying a different continuization for the second-order Heun's method, our approach implicitly utilizes the smoothness properties of $b(t,x)$ instead of the estimator $\hat{b}(t,x)$, and eventually reduces the smoothness requirement.
>
> To make these distinctions more explicit, we have added a new section in Appendix B that highlights the key novel techniques, explains how they differ from the approaches in these prior works, and clarifies why they yield improved bounds in our setting.
>
> **References**
>
> (Li et al., 2025b) Runjia Li, Qiwei Di, and Quanquan Gu. Unified convergence analysis for score-based diffusion models with deterministic samplers. In The Thirteenth International Conference on Learning Representations, 2025.
>
> (Huang et al., 2025) Daniel Zhengyu Huang, Jiaoyang Huang, and Zhengjiang Lin. Convergence analysis of probability flow ode for score-based generative models. IEEE Transactions on Information Theory, 2025.

---

> > ### Author Response · Authors · 2025-11-20
> >
> > ### Questions
> >
> > > In my understanding, many techniques are closely related and can be applied to previous analysis of diffusion ODEs. Do you think there are any properties specific to the stochastic interpolant problem (I.e., starting from another distribution instead of Gaussian noise)? For instance, can you provide some concrete examples to show the difference between the stochastic interpolant and the original diffusion process?
> >
> > A: Thank you for raising this important question. Yes, there are fundamental differences between the stochastic interpolant framework and traditional diffusion processes, particularly in their structural properties. A key distinction lies in the Markovian nature of the forward process.
> >
> > In standard variance-preserving diffusion models, the forward process is Markovian. For example, the SDE
> > $$\text{d}X_t=-X_t\text{d}t+\sqrt{2}\text{d}W_t,$$
> > admits an analytical transition kernel. Specifically, given $X_t$, the future state $X_{t+s}$ can be expressed as $$X_{t+s}=e^{-s}X_t+\sqrt{1-e^{-2s}}z,\quad z\sim\mathcal{N}(0,I_d),$$
> > which means the probability density $\rho_{X_{t+s}}(\cdot)$ is an explicit convolution of the scaled density $\rho_{e^{-s}X_t}(\cdot)$ and and a Gaussian. This Markov property can be useful in the analysis, for instance, when bounding the density ratios.
> >
> > In contrast, the stochastic interpolant $x_t=I(t,x_0,x_1)+\gamma(t)z$, is generally non-Markovian in $t$, because the interpolation $I$ depends explicitly on both $x_0$ and $x_1$. As a result, the relation between density of $x_{t+s}$ and $x_t$ are not simple as a convolutional update. This structural difference means that many analytical tools for diffusion process cannot be directly applied to the stochastic interpolant setting.
> >
> > We hope this answer addresses your question.
> >
> > -------
> >
> > We hope our response addresses your concerns. If so, we wonder if you could kindly consider raising your score? We will also be happy to answer any further questions you may have. Thank you very much!

---

> > > ### Comment · Reviewer_9Pnh · 2025-11-25
> > >
> > > Thank you for your response. The rebuttal has addressed my concerns, particularly the comparison between stochastic interpolants and diffusion models. Overall, I consider this a piece of good work. I will maintain my score and lean toward acceptance.
> > >
> > > Good luck!
> > > Reviewer 9Pnh

---

> > > > ### Author Response · Authors · 2025-11-25
> > > >
> > > > Thank you very much again for your thoughtful review and positive feedback! We truly value your advice. We look forward to the opportunity to address any further points or details you may have.

---

### Official Review · Reviewer_9RHo · 2025-10-30

**Soundness:** 3
**Presentation:** 3
**Contribution:** 3
**Rating:** 6
**Confidence:** 3

**Summary:**

This paper presents a theoretical study of iteration complexity for discretization schemes of stochastic-interpolant ODEs. As representative schemes, we analyze the first-order forward Euler method and the second-order Heun’s method. Under certain assumptions, complexity bounds for achieving $epsilon$-TV error for each scheme are derived. The analysis proceeds via carefully constructed continuous-time interpolants, which allow us to import continuous-time techniques to quantify discretization error and evaluate approximation accuracy.

**Strengths:**

**(S1) Important research theme.**
Investigating the iteration complexity of ODE solvers to ensure accurate generation with stochastic interpolants is an important research direction. Although diffusion models (in discretization form of neural ODE) are a major focus in machine learning, most evidence remains empirical. This work may provide the missing theoretical analyses underpinning diffusion models.

**(S2) Theoretical contributions.**
For general stochastic-interpolant ODEs, we derive the iteration complexity required to achieve $\epsilon$-TV error when using the first-order forward Euler method and the second-order Heun’s method.

**Weaknesses:**

**(W1) Positioning relative to prior work.**
In the Introduction (lines 048–049), you state that for ODE-based transformations, “the analysis has been limited to the continuous-time setting.” Meanwhile, Sec. 2.2 (lines 123–127) and Appendix B (Table 1) summarize prior results on the time needed for an ODE to reach $\epsilon$-TV error. Am I correct that all of these results are derived in continuous-time setting and do not account for discretization error? If YES, please revise Sec. 2.2 to make this explicit.

**(W2) Challenge level of the proposed analysis technique.**
My understanding is that the techniques in Sec. 4.1 (“Main ideas of the proof”)—which bridge the gap between a continuous-time process and a discrete-time estimator—are the key methodological contribution, enabling tools from continuous-time ODE analysis to be imported into the discretized setting. That said, it is somewhat difficult for me to gauge how technically challenging this bridge is. Could you additionally explain the main difficulties encountered in introducing this technique?

**Questions:**

(1) Definition of $d$ seems not appear in Introduction.

---

> ### Author Response · Authors · 2025-11-20
>
> Thank you for your time and effort in reviewing our paper! We are grateful for your constructive suggestions, which have significantly guided our improvements. Please find our responses to your comments below.
>
> > (W1) Positioning relative to prior work. In the Introduction (lines 048–049), you state that for ODE-based transformations, “the analysis has been limited to the continuous-time setting.” Meanwhile, Sec. 2.2 (lines 123–127) and Appendix B (Table 1) summarize prior results on the time needed for an ODE to reach epsilon-TV error. Am I correct that all of these results are derived in continuous-time setting and do not account for discretization error? If YES, please revise Sec. 2.2 to make this explicit.
>
> A: First, our work provides the first discrete-time analysis for stochastic interpolant ODEs, including the first error bounds for both Euler’s method and Heun’s method. As stated in the introduction, prior theoretical results for stochastic interpolant ODEs focus exclusively on continuous-time settings, leaving their discrete-time behavior unexplored.
>
> Because the formulation of stochastic interpolant ODEs closely resembles that of ODE-based diffusion models (e.g., DDIMs), and because diffusion models can be viewed within the broader stochastic interpolant framework, we consider it both natural and necessary to discuss and compare with existing analyses of diffusion ODEs. The results summarized in Table 1 (Appendix B) concern discrete-time analyses from the diffusion-model literature. The table serves to contextualize these prior works and to highlight how our new results extend them to the stochastic interpolant setting.
>
> In addition, we have included a new section in Appendix B that further elaborates on the main technical differences between our work and two of the most relevant prior analyses, clarifying the specific advances introduced by our approach.
>
> We hope this addresses your concerns satisfactorily.
>
> > (W2) Challenge level of the proposed analysis technique. My understanding is that the techniques in Sec. 4.1 (“Main ideas of the proof”)—which bridge the gap between a continuous-time process and a discrete-time estimator—are the key methodological contribution, enabling tools from continuous-time ODE analysis to be imported into the discretized setting. That said, it is somewhat difficult for me to gauge how technically challenging this bridge is. Could you additionally explain the main difficulties encountered in introducing this technique?
>
> A: As noted in our response to the previous question, we have included a new section in Appendix B that more thoroughly discusses our technical contributions relative to existing works. For both numerical methods, our proof strategy consists of two key components: (i) deriving an equivalent continuous-time formulation, and (ii) controlling the discrepancy between the true process and the estimated process.
>
> For the first component, the continuous-time formulation we use for Euler’s method aligns with prior work, as it is the most natural and standard approach. In contrast, our continuous-time formulation for Heun’s method differs from the existing literature, where we directly apply a linear approximation on the intermidiate drift $b(t,x)$, instead of performing a Taylor expansion on discrete timesteps. This new continuization plays an essential role in our analysis: it allows us to relax the required smoothness assumptions on the time derivative of $\hat{b}$. (See Appendix B.1 for details.)
>
> For the second component, establishing that the approximate drift $\tilde{b}(t,x)$ and its divergence remain close to the ground truth presents substantial technical challenges. Our approach introduces a structured sequence of approximations that decomposes the overall error into manageable parts, including (i) the estimation error from the difference between estimator $\hat{b}(t,x)$ and the real drift $b(t,x)$, (ii) bias error caused by the difference of $X_{t_k}$ and $z=F_{t_k\to t}^{-1}(X_t)$ or $G_{t_k\to t}^{-1}(X_t)$, as $z$ appears in the formulation of $\tilde{b}(t,X_t)$ instead of $X_{t_k}$, and (iii) the discretization error introduced by discrete-time approximation. By carefully bounding each term, where they are written as expectations, we obtain sharper error estimates than previous work.
>
> > Definition of $d$ seems not appear in Introduction.
>
> A: Thanks for pointing this out. The introduction is now slightly modified to include the definition of $d$.
>
> -------
>
> We hope our response addresses your concerns. If so, we wonder if you could kindly consider raising your score? We will also be happy to answer any further questions you may have. Thank you very much!

---

> > ### Comment · Reviewer_9RHo · 2025-11-25
> >
> > Thank you very much for your response. I have checked the additions in Appendix B, and my concerns (W1) and (W2) have been alleviated. (I had originally hoped for revisions to the main paper—for example, in the Introduction—but I think adding this material to Appendix B is also acceptable.)
> >
> > I will keep my score.

---

> > > ### Author Response · Authors · 2025-11-25
> > >
> > > Thank you very much again for your thoughtful review and positive feedback! We truly value your advice. We look forward to the opportunity to address any further points or details you may have.

---

### Official Review · Reviewer_Pr2N · 2025-10-31

**Soundness:** 3
**Presentation:** 3
**Contribution:** 3
**Rating:** 6
**Confidence:** 3

**Summary:**

This paper gives convergence analysis of ODE methods for stochastic interpolants under general conditions, in particular, the first-order (forward) Euler method and second-order Heun's method.

For Euler's method, the requirements are L^2 error bound on the drift $\epsilon_{drift}$, an expected Jacobian estimation error in Frobenius norm $\epsilon_{div}$, and smoothness bounds (Frobenius norm bound in the Jacobian & 2nd derivative of the score and Lipschitzness of divergence). The theorem gives $O(d^2/\epsilon)$ iteration complexity.

For Heun's method (with Jacobian estimation error in Frobenius norm squared), the theorem gives $O(d^{3/2}/\epsilon^{1/2})$ iteration complexity.

Experiments were run to obtain the empirical scaling of TV error with step size and dimension.

**Strengths:**

The paper is the first to give an analysis of the ODE discretization for stochastic interpolants under general conditions. This works for any interpolations with path $I$ and noise magnitude $\gamma(t)$ satisfying some smoothness conditions, and estimates satisfying the assumptions. Although the analysis techniques do not seem particularly novel (in being similar to previous ODE flow analyses), it is nevertheless valuable to work out the precise bounds in new setting under carefully detailed general assumptions, as in this paper. In particular, the theory shows the benefits of a higher-order ODE integrator and under what smoothness/error assumptions we can expect the improvement.

**Weaknesses:**

There is an additional factor of $d$ in Euler's method compared to diffusion models, and it is unknown whether this is a limitation of the analysis, or necessary in this general setting. (The experiments suggest the true scaling is linear.)

**Questions:**

It would be helpful to explain where the extra factor of $d$ come from compared to diffusion models.

* p. 6: "Divergencce" -> "Divergence"
* p. 8: "folloiwing" -> "following"
* p. 8: Missing period after "unbounded"

---

> ### Author Response · Authors · 2025-11-20
>
> Thank you for your time and effort in reviewing our paper! We are grateful for your constructive suggestions, which have significantly guided our improvements. Please find our responses to your comments below.
>
> > There is an additional factor $d$ of in Euler's method compared to diffusion models, and it is unknown whether this is a limitation of the analysis, or necessary in this general setting. (The experiments suggest the true scaling is linear.)
>
> > It would be helpful to explain where the extra factor of $d$ come from compared to diffusion models.
>
> A: Regarding the extra factor of $d$: in the case of variance-preserving (VP) diffusion models, the technique for removing this factor relies on a nontrivial result from stochastic localization. This result allows rewriting the expected score error as the derivative of a monotonic function. Without this specialized argument, the complexity bound for VP diffusion models under Euler’s method matches ours.
>
> However, this reduction crucially exploits the Gaussian-to-data geometry of VP diffusion and does not extend to general stochastic interpolants. At present, there is no analogous structural tool available in the general setting. Therefore, it remains unclear whether the additional $d$ factor is merely an artifact of analysis or fundamentally necessary. Reducing this factor is an open direction for future investigation.
>
> Finally, we emphasize that our work provides the first finite-time convergence analysis of stochastic interpolant ODEs under discrete-time ODE solvers, giving explicit error bounds for both Euler’s method (first order) and Heun’s method (second order), along with the resulting iteration complexities.
>
> > - p. 6: "Divergencce" -> "Divergence"
> > - p. 8: "folloiwing" -> "following"
> > - p. 8: Missing period after "unbounded"
>
> A: Thank you for pointing them out! We have fixed them in the revision.
>
> -------
>
> We hope our response addresses your concerns. If so, we wonder if you could kindly consider raising your score? We will also be happy to answer any further questions you may have. Thank you very much!

---

> > ### Comment · Reviewer_Pr2N · 2025-11-26
> >
> > I thank the authors for their response. I maintain my positive evaluation.

---

> > > ### Author Response · Authors · 2025-11-26
> > >
> > > Thank you very much again for your thoughtful review and positive feedback! We truly value your advice. We look forward to the opportunity to address any further points or details you may have.

---

### Official Review · Reviewer_TgjG · 2025-11-09

**Soundness:** 3
**Presentation:** 3
**Contribution:** 2
**Rating:** 4
**Confidence:** 3

**Summary:**

This paper investigates finite-time convergence of ODE samplers built from stochastic interpolants (SI). While previous studies have established finite-time bounds for SDEs, the ODE scenario has notably lacked discrete-time guarantees. The authors fill this gap by providing total variation (TV) distance error bounds and detailing the iteration complexity for both the forward Euler and Heun's methods under regularity and estimator-error assumptions.

**Strengths:**

- The paper addresses a significant theoretical gap for SI-based ODEs by providing finite-time TV bounds and iteration complexities for both Euler and Heun methods within the SI framework.
- Mathematical proofs are solid and clear. Discrete-to-continuous interpolation yields a piecewise ODE, enabling drift/divergence comparisons to bound TV.
- 2D tasks and Gaussian-mixture tests verify $O(h)$ (Euler) and $O(h^2)$ (Heun) discretization orders.

**Weaknesses:**

- The requirement of uniform Lipschitz on $\hat{b}$ and its divergence (Assumption 4.4) seems kind of idealized. Is it possible to provide an empirical verification?
- Experiment demos are limited to three 2D transformations and d-dim Gaussian mixtures; Is it possible to provide results on real-data benchmarks?
- The theory in this paper establishes error bounds that scale with $d^2$ and $d^3$. However, the empirical findings indicate a roughly linear correlation, suggesting room for further refining on the bounds.

**Questions:**

See the weaknesses.

---

> ### Author Response · Authors · 2025-11-20
>
> Thank you for your time and effort in reviewing our paper! We are grateful for your constructive suggestions, which have significantly guided our improvements. Please find our responses to your comments below.
>
> >The requirement of uniform Lipschitz on $\hat{b}$ and its divergence (Assumption 4.4) seems kind of idealized. Is it possible to provide an empirical verification?
>
> A: Thank you for your insightful comment.
>
> First, it is important to note that our work presents the first discrete-time analysis of stochastic interpolant ODEs, providing novel error bounds for both the first-order Euler method and the second-order Heun method. These results, to the best of our knowledge, are unique in this context.
>
> Regarding the concern about the uniform Lipschitz assumption on the drift $\hat{b}$ and its divergence (Assumption 4.4), we acknowledge that this may appear idealized at first glance. However, the Lipschitz assumption is commonly used in theoretical works on diffusion ODEs (e.g. [1,2,3]), where it is essential for controlling the overall distribution error.
>
> Additionally, we demonstrate in Lemma E.7, that under the condition that the data has bounded support (as is typically the case in datasets like images), the true drift $b(t,x)$ and its divergence are indeed Lipschitz continuous. This provides a concrete foundation for the assumption in practical settings, where such conditions are often met in real-world data.
>
> **References**
>
> [1] Daniel Zhengyu Huang, Jiaoyang Huang, and Zhengjiang Lin. Convergence analysis of probability flow ode for score-based generative models. IEEE Transactions on Information Theory, 2025.
>
> [2] Joe Benton, George Deligiannidis, and Arnaud Doucet. Error bounds for flow matching methods. Transactions on Machine Learning Research, 2024.
>
> [3] Runjia Li, Qiwei Di, and Quanquan Gu. Unified convergence analysis for score-based diffusion models with deterministic samplers. In The Thirteenth International Conference on Learning Representations, 2025.
>
> >Experiment demos are limited to three 2D transformations and d-dim Gaussian mixtures; Is it possible to provide results on real-data benchmarks?
>
> A: Thank you for the suggestion. We have included a new experiment on image-generation tasks, as described in Appendix C. Specifically, we adopt the Gaussian-to-image setting discussed in our complexity analysis. In this setup, the drift function $b(t,x)$ can be expressed as a function of $x$ and the score $s(t,x)$, which allows us to leverage pretrained score networks to evaluate different ODE solvers.
>
> As shown in Figures 3 and 4 in Appendix C, the second-order Heun method exhibits faster convergence than the first-order Euler method, consistent with our theoretical predictions.
>
> We hope these new results adequately address your concern.
>
> > The theory in this paper establishes error bounds that scale with $O(d^2)$ and $O(d^3)$. However, the empirical findings indicate a roughly linear correlation, suggesting room for further refining on the bounds.
>
> A: We appreciate the reviewer’s observation about the empirical findings suggesting a linear correlation. It is important to note that our work presents the first convergence analysis for ODE-based stochastic interpolants, and optimizing the error bounds remains an interesting open research question.
>
> While a nearly $O(d)$ convergence rate has been established for the variance-preserving diffusion model using a first-order ODE solver, the analysis there relies on stochastic localization techniques that are heavily dependent on the Gaussian structure. To date, no method exists that generalizes these results to the broader class of stochastic interpolants considered in our work.
>
> Furthermore, we agree that refining the $d$-dependence in the error bound for higher-order ODE solvers is an exciting avenue for future research, both for diffusion models and stochastic interpolants in general. This is an active area of investigation, and we hope our results will serve as a foundation for further improvements in this direction.
>
> -------
>
> We hope our response addresses your concerns. If so, we wonder if you could kindly consider raising your score? We will also be happy to answer any further questions you may have. Thank you very much!

---

> ### Author Response · Authors · 2025-11-26
>
> Dear Reviewer,
>
> We greatly appreciate the effort and expertise you have contributed to reviewing our paper. Since the author reviewer discussion period is ending soon, we hope that our responses have addressed your concerns. If that is the case, we wonder if you could kindly consider raising your score rating? Should you have any additional questions, we are more than happy to provide further assistance. Thank you very much for the support!

---

### Author Response · Authors · 2025-11-20

Dear ACs and reviewers,

Thank you for your time and effort in reviewing our paper! We are grateful for your constructive suggestions, which have significantly guided our improvements.

We have revised our paper according to the review comments, and the updated parts are highlighted in red in our revision. The main changes are as follows:

- We have added a new Section B.1 in the Appendix, which compares our technical details with two closely related previous works.
- We have added a new Section C in the Appendix, which presents additional experiments on image generation.
- We have provided more detailed explanations in the proofs to improve clarity.
- We have corrected typos and made additional modifications to align with the above changes.
- We have refined the paper’s presentation to enhance clarity and improve the overall flow.


We hope our response addresses your concerns. If so, we wonder if you could kindly consider raising your score? We will also be happy to answer any further questions you may have. Thank you very much!

Best regards,
Authors

---

### Meta-Review · Area_Chair_7eN3 · 2025-12-24

**Summary:**

Reviewers raised concerns about the strength of the assumptions (e.g., assumptions on the boundedness of errors in estimating the derivative of the score), and the quantitative bounds which incur worse dimension dependence than the previous works on diffusion models. However, the setting of stochastic interpolants and flow-based models seems to be more challenging, and in that regard, this paper is a solid theoretical contribution. The authors did a good job at responding to these concerns; in particular, from my own understanding, I believe that the weaknesses are consistent with known technical obstructions.

Reviewer TgjG also brought up experimental evaluation. As the contribution of this paper is purely theoretical, I believe this concern should be disregarded.

The scores are somewhat borderline, but this paper meets the bar.

**Reviewer Concerns:**

The concerns regarding the theory, while not fully resolved (poor dimension dependence, etc.), are challenging, and it would be unrealistic to expect the results to be significantly improved in a short time frame.

As noted, I believe that the concern regarding experimental validation is unfounded. Stochastic interpolants are widely used in practice and there are plenty of empirical works on their performance. The contribution of this paper is theoretical.

**Reviewer Scores:**

Reviewers Pr2N, 9RHo, and 9Pnh were satisfied by the rebuttal and maintained their scores.

Given the technical discussions that the authors had with the other reviewers, it is possible that reviewer TgjG would have slightly raised the score.

---

### Decision · Program_Chairs · 2026-01-26

Accept (Poster)